# Early visual signatures and benefits of intra-saccadic motion streaks

**Richard Schweitzer** [1,2,3¤*], **Thomas Seel**[4], **Jörg Raisch**[3,5], **Martin Rolfs**[2,3,6]

**1** Centro Interdipartimentale di Mente e Cervello, Università degli studi di Trento, Trento, Italy, **2** Department of Psychology, Humboldt-Universität zu Berlin, Berlin, Germany, **3** Science of Intelligence, Research Cluster of Excellence, Berlin, Germany, **4** Institute of Mechatronic Systems, Leibniz Universität Hannover, Hannover, Germany, **5** Control Systems Group, Technische Universität Berlin, Berlin, Germany, **6** Bernstein Center for Computational Neuroscience Berlin, Berlin, Germany

¤ Centro Interdipartimentale di Mente e Cervello, Palazzo Fedrigotti, Corso Bettini 31, Rovereto (TN), Italy
* richard.schweitzer@unitn.it

**Data availability statement:** Except for modeling, all data analyses performed were planned and pre-registered, unless otherwise

## Abstract

Eye movements routinely induce motion streaks as they shift visual projections across the retina at high speeds. To investigate the visual consequences of intra-saccadic motion streaks, we co-registered eye tracking and EEG while gaze-contingently shifting target objects during saccades, presenting either continuous, 'streaky' or apparent, step-like motion in four directions. We found significant reductions of secondary saccade latency, as well as improved decoding of the post-saccadic target location from the EEG signal when motion streaks were available. These signals arose as early as 50 ms after saccade offset and had a clear occipital topography. Using a physiologically plausible visual processing model, we provide evidence that the target's motion trajectory is coded in orientation-selective channels and that speed of gaze correction was linked to the visual dynamics arising from the combination of saccadic and target motion, providing a parsimonious explanation of the behavioral benefits of intra-saccadic motion streaks.

## Author summary

To efficiently explore their visual environment, humans incessantly make brief and rapid eye movements. These saccades inevitably shift the entire visual image across the retina, thereby inducing–like a moving camera with long exposure duration–a significant amount of motion blur, transforming single objects into elongated smeared motion streaks. While simultaneously recording electroencephalography and eye tracking, we asked human observers to make saccades to a target stimulus which then rapidly changed location while their eyes were in mid-flight. Critically, we compared smooth target motion to a simple jump, thus isolating neural responses and behavioral benefits specific to motion streaks: For continuous motion (i.e., when streaks were available), the post-saccadic target location could be decoded earlier from electrophysiological data and

stated. Pre-registration, data, and analysis scripts are made publicly available and can be accessed at https://osf.io/qm5ca/ (https://doi.org/10.17605/OSF.IO/QM5CA). All modeling code can be found at https://github.com/richardschweitzer/simplified_V1 and has been archived on Zenodo (https://doi.org/10.5281/zenodo.15837947). Modeling results are made available at https://osf.io/7phg4/ (https://doi.org/10.17605/OSF.IO/7PHG4).

**Funding:** R.S., M.R., and J.R. were funded by the Deutsche Forschungsgemeinschaft (DFG, German Research Foundation) under Germany's Excellence Strategy - EXC 2002/1 "Science of Intelligence" - project number 390523135. R.S. was supported by the Studienstiftung des deutschen Volkes during the early stages of the study, as well as by the Italian Ministero dell'Università e della Ricerca (grant 'T-GAZE', MSCA_0000027_FIS02) in the final stages of the study. M.R. has received funding from the European Research Council (ERC) under the European Union's Horizon 2020 research and innovation programme (grant agreement No 865715) as well as from the Heisenberg Programme of the DFG (grants RO3579/8-1 and RO3579/12-1). The funders had no role in study design, data collection and analysis, decision to publish, or preparation of the manuscript.

**Competing interests:** The authors have declared that no competing interests exist.

secondary saccades went more quickly to the new target location. Indeed, decoding of target location succeeded immediately after the end of the saccade and was most efficient on occipital sensors, suggesting that saccade-induced motion streaks are represented in visual cortex. Computational modeling of saccades as a consequence of early visual processes suggests that fast motion could be efficiently coded in orientation-selective channels, providing a parsimonious mechanism by which the brain exploits motion streaks for goal-directed behavior.

## 1. Introduction

When objects in our environment move, they shift their projection across the retina and, due to visual system's properties of temporal summation and persistence, may well induce motion blur. Whereas, at moderate velocities, sufficient presentation durations may suffice to reduce the impression of smear [1], very high velocities–such as those that routinely occur during rapid eye movements (i.e., saccades with peak speeds of more than 400 deg/s; [2])–inevitably induce visible smear. These so-called intra-saccadic motion streaks can be readily perceived as smeared elongated traces when making saccades across high-contrast and spatially circumscribed objects in sparse environments, such single or multiple light sources in a dimly lit room (e.g., [3–8]). In contrast, they are rarely detected in more natural, everyday vision–in fact, human observers remain largely unable to discriminate even large-field intra-saccadic stimulation, provided it is sandwiched between stable pre- and post-saccadic images [9,10], a phenomenon often attributed to forward and backward masking [11–13]. A similar principle may apply to objects moving at saccadic speeds: Human observers performed close to chance level when their task was to explicitly match pre- and post-saccadic object locations based on streaky high-speed motion presented strictly during saccades [14]. Performance however recovered to fixation level when a brief blanking interval around saccade offset was introduced, suggesting that object motion during saccades was not impossible to process, but temporally masked by the post-saccadic presence of the object (see [5,15], for similar arguments). Despite this efficient impairment of perception of motion smear due to masking, intra-saccadic motion streaks still serve as a visual cue that guides oculomotor behavior: Streaks congruent with post-saccadic target location and identity improved the accuracy and latency of secondary saccades, whereas incongruent streaks had the opposite effect [16]. The idea that emerged from these findings, that is, that spatiotemporal continuity across saccades could facilitate tracking object location by establishing correspondence between pre- and post-saccadic object locations, may make intuitive sense, but lacks in explanatory power because the implementation of this putative mechanism remains unclear: What information could constitute this continuity and what is the locus and time course of the facilitation of secondary saccades? Or, put differently, how can high-speed object motion during saccades be represented in the brain, so that the system can make use of it? Note that this first research question bears a second, broader research question that refers to one of the major challenges of visual stability, that is, the omission of motion smear: How can the visual system use intra-saccadic signals to inform oculomotor processes while otherwise routinely omitting saccade-induced visual transients from perception?

Addressing the first research question, we co-registered eye tracking and electroencephalography (EEG) to shed light on the neuronal signatures of intra-saccadic motion streaks and understand how they might be used for the facilitation of secondary saccades. We approached this question on three different levels which will be presented sequentially in this paper. First, on the behavioral level, we inspected the conditions under which motion streaks

reduce the latency of secondary saccades. Second, on the physiological level, we applied multivariate pattern analysis (MVPA) to EEG data to decode both the presence of strictly intra-saccadic streaky object motion presented with a frame rate of 1440 fps and the location of a target stimulus as it is rapidly displaced during saccades. We found that continuous object motion during saccades induced reliable neural responses as early as 50-60 ms after saccade offset and with clear occipital topography, which allowed post-saccadic target locations to be decoded earlier in time. Third, on the computational level, we showed how these responses can arise with the help of a model mimicking the spatial and temporal response characteristics of the early visual system. Feeding it information about target stimuli and their retinal trajectories, its output revealed that experimental results could well be explained in terms of the coding of fast motion in orientation-selective channels. Finally, in a first attempt to address the second research question, we devised an integrative framework to describe how the system could actively monitor and selectively use intra-saccadic visual transients. As a proof of plausibility, we used this framework to predict both absolute secondary saccade latencies and motion-induced latency benefits, providing a direct link between the early visual dynamics of the stimulus and the speed of gaze correction.

## 2. Results

To understand how the visual system processes intra-saccadic motion streaks, we used a paradigm (Fig 1a) that would allow contrasting continuous object motion with mere apparent object motion [14,16]. Human observers made large horizontal saccades (primary saccades) of 17.6 deg mean amplitude (range [15.5, 19.6]), 53.6 ms duration (range [45.5, 67.8]), and peak velocities of 528.2 degrees of visual angle per second (dva/s; range [410.4, 644.0], see also S5 Figa and S5 Figb and S5 Figc) towards a noise patch stimulus bandpass-filtered to low spatial frequencies (SFs, see Sect 5.4), henceforth, the target. Once the onset of the primary saccade was detected, the target either remained at its initial location or moved rapidly in one of four cardinal directions, traveling a distance of 6.6 dva in only 25 ms and strictly during the primary saccade (see Sect 5.5). Importantly, target motion occurred in either a continuous (motion-present) or apparent (motion-absent) fashion, thereby creating a motion streak or leaving a blank during the motion interval, respectively (Fig 1b). After the offset of the primary saccade, observers made a secondary saccade to the (by then) displaced target.

### 2.1. Impact of intra-saccadic target motion on secondary-saccade latency

Behaviorally, we had previously shown that secondary saccades are initiated faster when intra-saccadic motion streaks are present and consistent with the post-saccadic location of a target [16]. We thus first ascertained that we could replicate the facilitating effects of continuous motion on secondary saccade latency in this slightly altered version of the paradigm. To perform this planned and pre-registered analysis of secondary saccades in motion-absent and motion-present conditions, we selected those trials in which primary and secondary saccades were correctly performed and landed in the direct vicinity of the target (see Sect 5.7.1). Secondary saccades in static conditions were included, as in the absence of target displacement they may naturally occur as corrective saccades [17,18]. With a static target, secondary saccades were made in 65.6% (range [42.2, 92.7]) and 65.5% (range [41.4, 92.0]) of all trials, in present and absent conditions, respectively. Of those secondary saccades 98.3% (range [91.2, 100]) and 98.2% (range [89.8, 100]) were aimed at the target. As a comparison, with moving targets secondary saccades were made in 98.9% (range [97.3, 100.0]) and 99.0% (range [98.1, 100.0]) of trials, of which 98.4% (range [95.9, 99.8]) and 98.6% (range [97.4, 99.7]) were

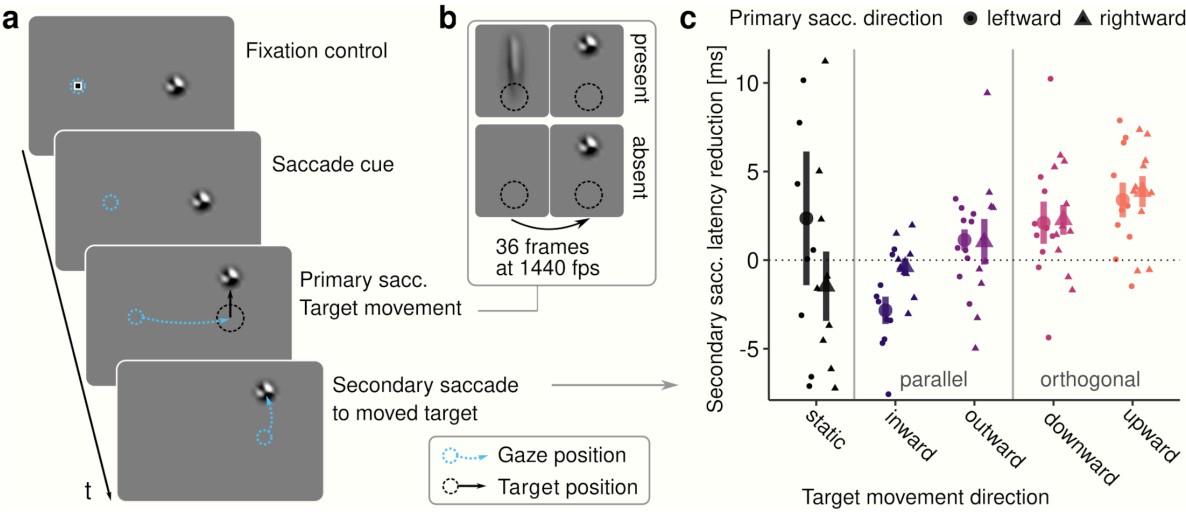

**Fig 1. Experimental paradigm. a** Upon the disappearance of a fixation marker human observers made a primary saccade towards a noise patch stimulus, the target (see Sect 5.4). During the primary saccade the target rapidly moved in one of the four cardinal directions, or not at all (static condition). Once landed, observers initiated a secondary saccade towards the target's new location. **b** The target movement occurred either as a continuous and linear motion (present condition) or as a step with a blank interval between the trajectory's endpoints (absent condition). In both conditions, the movement had a duration of 25 ms and was initiated and concluded strictly during the primary saccade (see S5 Figd and S5 Fige). **c** Differences in secondary saccade latency (reduction computed as absent – present condition) depending on the saccade's and the target's movement directions. Small points indicate differences of individual observers, large points show population means ± one standard error of the mean (SEM).

target-bound. This data suggests that observers made highly accurate secondary saccades regardless of the intra-saccadic motion condition they were subjected to.

Secondary saccade latency, here defined as the time elapsed between primary saccade offset and secondary saccade onset, varied considerably across target movement directions ($F(4, 36) = 30.59$, $\eta^2 = 0.552$, $p < .001$, $p_{GG} < .001$, see S6 Fig for observer distributions): Latencies in response to outward displacements were shortest ($M = 123.8$ ms, $SD = 11.1$), whereas inward displacements produced the longest latencies ($M = 167.3$ ms, $SD = 27.6$). Furthermore, upward displacements ($M = 134.2$ ms, $SD = 15.6$) were corrected faster than downward displacements ($M = 155.0$ ms, $SD = 21.7$). The direction of the primary saccade (leftward or rightward) did not influence secondary saccade latencies ($F(1, 9) = 0.71$, $\eta^2 = 0.002$, $p = .419$), and did not modulate the effect of target movement direction ($F(4, 36) = 0.67$, $\eta^2 = 0.001$, $p = .614$, $p_{GG} = .491$). Importantly, we observed a small, but significant main effect of continuous motion ($F(1, 9) = 8.81$, $\eta^2 < 0.001$, $p = .016$), as well as a significant interaction with target movement direction ($F(4, 36) = 3.59$, $\eta^2 = 0.001$, $p = .014$, $p_{GG} = .044$), suggesting that continuous motion affected secondary saccades, but not in the same way across movement directions. To better understand this interaction between the presence of continuous motion and the target's movement direction, we subtracted latencies in the present condition from those in the absent condition, so that positive differences would indicate a benefit of continuous motion (Fig 1c). Although we observed a significant net benefit across all conditions ($M = 1.2$, $SD = 5.2$), as indicated by the significant main effect above, a significant benefit applied only to target movement directions orthogonal to the saccade direction, that is, downward ($M = 2.2$, $SD = 2.7$, $t(9) = 2.53$, $p = .032$) and upward ($M = 3.6$, $SD = 2.5$, $t(9) = 4.47$, $p = .002$) motion. Continuous target movement parallel and in the direction of the saccade (outward condition) did not produce a significant latency benefit ($M = 1.1$, $SD = 1.9$, $t(9) = $

1.81, $p$ = .104), and target movement against the direction of the saccade (inward condition) even resulted in a slight increase of saccade latency ($M$ = –1.6, $SD$ = 1.7, $t(9)$ = –2.96, $p$ = .016), suggesting that under some circumstances continuous target motion could also induce a cost for saccade initiation. When targets remained at their original location, continuous motion did not affect the latency of eventual secondary saccades ($M$ = 0.4, $SD$ = 5.4, $t(9)$ = 0.26, $p$ = .803). In this condition, however, latency distributions were much broader than those observed with target movement (S6 Fig). Note that, while target movement direction played a large role in modulating the secondary saccades, primary saccade direction did not further modulate these relationships ($F(4, 36)$ = 0.95, $\eta^2$ = 0.044, $p$ = .446, $p_{GG}$ = .378). We also found no evidence that systematic differences in the metrics of primary saccades or stimulus timing could be responsible for the results above (S7 Fig and S8 Fig), except for an extremely small but systematic earlier stimulus onset in motion-absent condition ($M_{absent}$ = 17.4, $M_{present}$ = 17.5, $F(1, 9)$ = 18.87, $\eta^2$ = 0.007, $p$ = .001), likely due to faster drawing to the graphics card's back buffer when the target was not shown. However, this difference not only was in the sub-millisecond range, but also would have only contributed to a dampening of the latency effect due to an earlier onset of the post-saccadic target location. To sum up, the reductions of secondary saccade latencies in response to intra-saccadic continuous target motion orthogonal to the primary saccade's direction replicate our previous findings [16]. This effect did not generalize to target motion parallel to the direction of the saccade—a result reminiscent of saccadic suppression of displacement, which is equally specific to target displacements along the direction of the saccade [19].

## 2.2. Decoding of the EEG signal

Having established that intra-saccadic continuous motion is visually processed and used for gaze correction, we hypothesized (see pre-registered predictions at https://osf.io/2vmpr/) that the physiological correlates of continuous target motion should be present in the visual signal even before the secondary saccade is initiated. Within the bounds of our experimental design, time-resolved MVPA may be used to perform three major tests. First, in the context of the behavioral facilitation of secondary saccades, it is of major interest at what point in time the target's movement direction–and thus its post-saccadic location–can be decoded: If intra-saccadic motion streaks have indeed the ability to link pre- and post-saccadic object locations (as suggested in [14,16]), we would expect that they render the target's displacement available earlier than if they were absent. Second, it should be possible to decode from the EEG signal whether or not continuous motion has been presented in a given trial, providing insights in the immediate visual consequences of our stimulus and processing time course. Third, resolving not only the temporal but also spatial patterns of decoding accuracy sheds light on which (approximate) brain regions are involved when observers perform the gaze-correction task at hand.

**2.2.1. Decoding post-saccadic target location.** Addressing the results of the first test, the decoding of the target's movement direction, or its post-saccadic target location, Fig 2b shows the time course of classification accuracy relative to the offset of the primary saccade. We chose this reference point not only because the offset of the target's intra-saccadic movement was very close to saccade offset (S5 Fige and S7 Fig, rightmost column), but also because the earliest fixation-related visual potential, the $\lambda$ wave, is often shown as locked to saccade offset [20–22]. Note that the overall pattern of results did not change when decoding analyses were performed relative to target movement offset, instead of primary saccade offset (compare Fig 2b and 2c with S10 Figa and S10 Figb). Clearly, the EEG signal carried information about the target's motion direction: Classification accuracy increased sharply around 50 ms

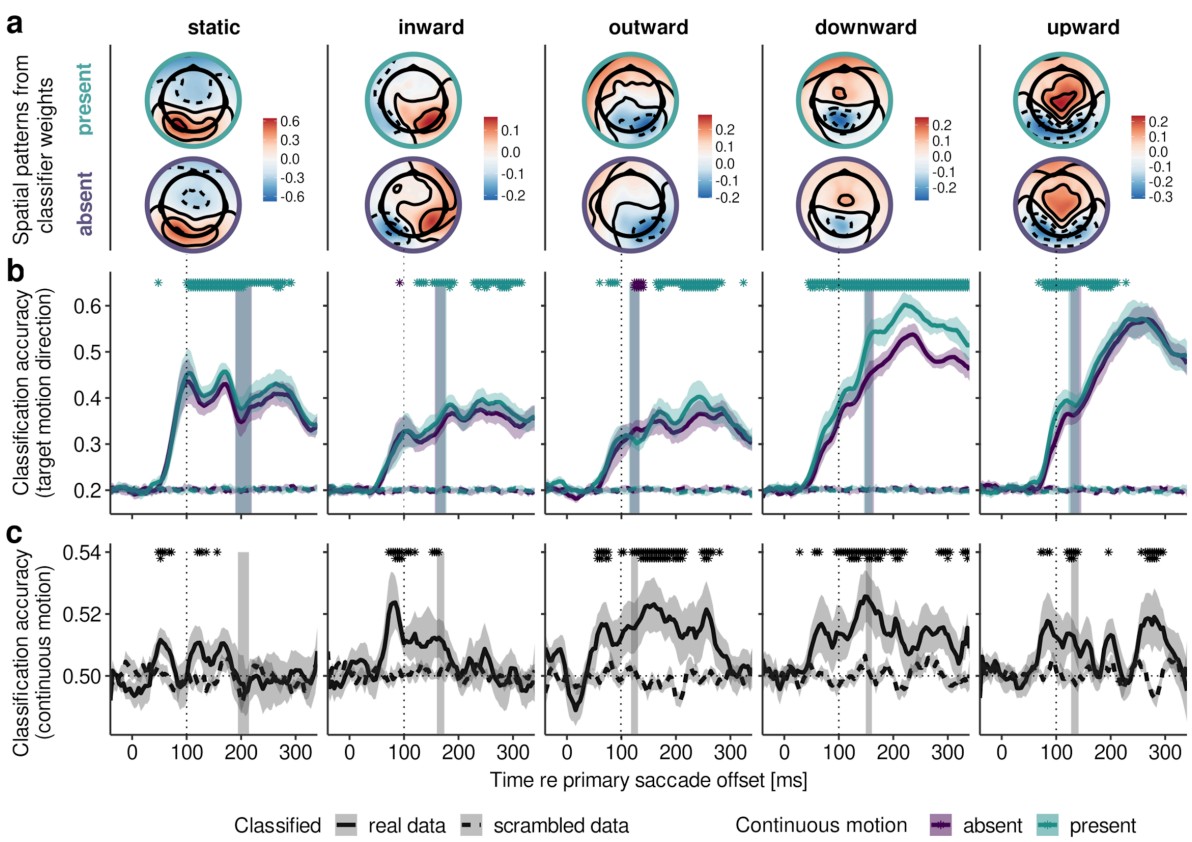

**Fig 2. EEG decoding results for each target motion direction. a** Spatial patterns [a.u.] for present and absent conditions (upper and lower row, respectively) computed from all channels' classifier weights (decoding target motion direction in a one-versus-rest fashion) at the time of 100 ms after saccade offset. Patterns were averaged across primary saccade direction, horizontally flipping channel locations to adjust for retinotopy. **b** Accuracy of classifying target motion direction over time after primary saccade offset, separately for present and absent conditions. Asterisks indicate results of cluster-based permutation tests comparing these two conditions, and shaded vertical areas indicate 95% confidence intervals for the mean onset time of secondary saccades. **c** Accuracy of classifying continuous motion, that is, present vs absent conditions. To determine whether classification accuracy was above chance level, cluster-based permutation tests compared classification performance for real and scrambled class labels. All shaded error bars indicate ±1 SEM. First-row asterisks denote a significance level of <.05, whereas second-row denote <.01. All dashed lines show baseline classification accuracy, computed by performing the decoding procedure with scrambled class labels.

after saccade offset, reaching a first peak at around 100 ms–the typical latency of the occipital $\lambda$ response occurring after real and simulated eye movements [21,23–26], which was in the same way present in our data after both primary and secondary saccades (S4 Fig). Inspecting the spatial patterns in Fig 2a, it is evident that the classifier, too, relied primarily on occipital electrodes. Patterns also seemed to reflect the (inverted) expected cortical locus of the target's position in the visual field, for instance, stronger parietal topography for downward and stronger occipital topography for upward target motion. While the first peak at 100 ms was especially prominent when the target remains in the same location, the overall classification accuracy varied considerably across target motion directions (Fig 2b): With reference to static targets (mean maximum classification accuracy; $M = 0.53$, $SD = 0.10$, with a chance level at 0.2), target motion orthogonal to the saccade's direction was classified particularly well, especially after secondary saccades were executed (downward: $M = 0.61$, $SD = 0.06$; upward: $M = 0.61$, $SD = 0.08$), whereas target motion parallel to the saccade's direction was classified less accurately in comparison (inward: $M = 0.45$, $SD = 0.07$; outward: $M = 0.43$, $SD = 0.10$)–a

highly significant effect across the sample ($F(4, 36) = 27.47$, $\eta^2 = 0.460$, $p < .001$, $p_{GG} < .001$). Note that this difference in classification accuracy does not reflect a difference in oculomotor behavior, as for non-static targets only trials with target-bound secondary saccades were included in the decoding analysis. For the sake of completeness, there was a marginally significant difference in secondary saccade rates across target motion directions ($F(3, 27) = 4.29$, $\eta^2 = 0.242$, $p = .013$, $p_{GG} = .049$), but this mean difference amounted to 1.5% at most (mean secondary saccade rates; inward: 98.0%, SD = 1.7; outward: 99.7%, SD = 0.5; downward: 98.6%, SD = 1.5; upward: 99.5%, SD = 0.6), making it extremely unlikely that variation in secondary saccade rate could have accounted for the differences in classification accuracy. It is equally unlikely that the difference between target motion conditions was due to a poor removal of corneoretinal artifacts in vertical saccades, as the procedure applied to remove eye movement-related artifacts (see Sect 5.7.2) worked equally well for all secondary saccade directions (S3 Fig). Finally, including only time points prior to each observer's mean secondary saccade onset in a given condition, we found that the same reliable difference between parallel and orthogonal target motion already emerged before secondary saccades were performed (mean maximum classification accuracy; parallel: $M = 0.36$, SD = 0.08; orthogonal: $M = 0.44$, SD = 0.07; $F(1, 9) = 11.84$, $\eta^2 = 0.224$, $p = .007$). In short, even though secondary saccades were executed timely and correctly across all target motion directions, we find that displacements parallel to the primary saccade's direction were harder to decode from the EEG signal than those orthogonal to it.

In agreement with our hypothesis regarding an earlier availability of the target's post-saccadic location, continuous motion led to the improved classification of motion direction across conditions and time points (mean maximum classification accuracy; present: $M = 0.54$, SD = 0.08; absent: $M = 0.51$, SD = 0.07; $F(1, 9) = 68.32$, $\eta^2 = 0.026$, $p < .001$), but to varying degrees for different target motion directions ($F(4, 36) = 7.84$, $\eta^2 = 0.021$, $p < .001$, $p_{GG} = .001$). Continuous motion improved decoding accuracy as early as 52 ms and 76 ms after primary saccade offset in the downward and upward conditions, 60 ms and 124 ms in the outward and inward conditions, and 100 ms in the static condition (Fig 2b). Focusing again on the critical interval prior to the execution of secondary saccades, analyses of classification accuracy revealed both a significant main effect ($F(1, 9) = 23.51$, $\eta^2 = 0.016$, $p < .001$) and a significant interaction between target movement and presence of continuous motion ($F(4, 36) = 3.56$, $\eta^2 = 0.016$, $p = .015$, $p_{GG} = .039$), confirming that continuous target motion reliably improved decoding of displacement direction even before secondary saccades were performed, but not equally across motion direction. In order to make sure that low-latency secondary saccades–which may well have occurred earlier than each observer's estimated mean secondary saccade latency–were not the primary cause of this improvement, we excluded all trials with secondary saccade latencies below 100 ms, that is, on average 3.8% of trials (SD = 4.7), and reran the decoding of post-saccadic target location. Results of this control analysis were virtually identical to those reported above (for a comparison, see S11 Fig), confirming that the earliest improvements in decoding accuracy were not merely the result of early eye movements. The strongest benefits in classification accuracy were achieved for downward ($M = +6.25\%$, SD = 4.64) and upward ($M = +2.61\%$, SD = 2.32) motion, whereas hardly any benefits were found for inward ($M = +1.27\%$, SD = 4.98) and outward ($M = −0.81\%$, SD = 5.19) target motion, or static targets ($M = +2.30\%$, SD = 2.68). This replicated the qualitative difference between parallel and orthogonal target motion directions and showed that the effect of continuous motion was strongest for orthogonal target motion ($M = +4.43\%$, $t(9) = 5.66$, $p < .001$), reduced for static targets ($M = +2.31\%$, $t(9) = 2.72$, $p = .023$), and absent for parallel target motion ($M = +0.23\%$, $t(9) = 0.27$, $p = .786$). Remarkably, this pattern of increasing decoding accuracy is similar to the patterns of decrease in secondary saccade latency due

to continuous motion: For both dependent variables the effect was exclusive to orthogonal target motion directions, suggesting a tight link between the two.

**2.2.2. Generalization from apparent to continuous motion.** The early onset and improvement of decoding accuracy arising from the presentation of continuous (as opposed to step-like) target motion during saccades suggests that these signals were not preemptively removed from visual processing, for instance, as early as in the LGN [27], but that they elicited unique cortical activation patterns that classifiers were well able to extract. It is however less straightforward to answer whether these patterns were systematically informative about the post-saccadic location of the target. In fact, it could be argued that it would only be natural to find earlier decoding of the target's location in the case of continuous motion: Since motion was always contingent upon the to-be-decoded post-saccadic target location, classifiers could have learned from the neural responses to the target motion presented during the 25-ms interval, whereas in the motion-absent condition they could not. To rule out the possibility that the systematic differences between motion-present and motion-absent conditions were merely an artifact of classifying post-saccadic target location separately in these conditions, we devised a generalization test in which we trained classifiers exclusively on motion-absent trials, so that they could not have picked up the target's streaky motion specifically, and then let them predict post-saccadic target location in motion-present trials. As shown in S9 Fig, classifiers trained exclusively on motion-absent trials exhibited improved accuracy when predicting motion-present trials ($M = 0.530$, $SD = 0.070$) compared to when predicting motion-absent trials ($M = 0.514$, $SD = 0.068$; $F(1, 9) = 66.81$, $\eta^2 = 0.009$, $p < .001$), and again this improvement did not occur similarly across all target motion directions ($F(2, 18) = 4.12$, $\eta^2 = 0.004$, $p = .035$, $p_{GG} = .044$). Before secondary saccades were made, classification of target movement orthogonal to primary saccade direction was significantly improved ($M = +1.9\%$, $t(9) = 3.00$, $p = .015$), classification of static targets improved with only marginal significance ($M = +2.1\%$, $t(9) = 2.22$, $p = .053$), whereas parallel target movement clearly did not elicit any improvement ($M = -0.6\%$, $t(9) = -1.19$, $p = .261$). The generalization test thus confirmed results up to this point–despite the classifiers' inability to learn from continuous target motion. This suggests that early-onset, improved classification accuracy was not an artifact of the stimulus or classification procedure used, but was most likely related to a strengthened direction signal: To some extent, the visual signal resulting from continuous motion must have given away the post-saccadic target location, allowing the classifier to extract this pattern earlier than it would have otherwise been able to.

**2.2.3. Decoding intra-saccadic continuous motion.** Results up to this point indicate that intra-saccadic motion signals not only lead to earlier classification of post-saccadic object locations, but also to reduced latencies of secondary saccades made to these object locations. The second test we planned to perform is whether the presence of these intra-saccadic motion signals themselves can also be decoded from EEG signals. In this analysis the classifier determines for any given target motion direction whether continuous or apparent motion was presented. Due to the brevity and velocity of moving targets, as well as the fact that streaks were not actively attended and post-saccadically masked [5,14,28,29], this question is not trivial. Indeed, the accuracy of classification of target-present versus target-absent conditions was very low, rarely exceeding 52% (chance level: 50%; Fig 2c), suggesting a weak signal compared to classification of post-saccadic target location. Despite its low absolute value, classification accuracy was significantly above chance, irrespective of target motion direction and before secondary saccades were made. Accuracy deviated from chance level (determined empirically by scrambling class labels) as early as 48 ms in static (mean classification accuracy: $M = 50.9\%$, $SE = 0.4$, $p = .029$), 72 ms in inward ($M = 51.9\%$, $SE = 1.08$, $p = .025$), 56 ms in outward

($M$ = 51.3%, $SE$ = 0.66, $p$ = .011), 56 ms in downward ($M$ = 51.3%, $SE$ = 0.71, $p$ = .029), and 72 ms upward ($M$ = 51.2%, $SE$ = 0.74, $p$ = .043) conditions. Two aspects of these results are particularly interesting. For one, above-chance classification arose less than 100 ms after saccade offset and long before secondary saccades were made (mean maximum classification accuracy: 53.1%; deviation from scrambled data: $F(1, 9)$ = 14.8, $\eta^2$ = 0.48, $p$ = .004), that is, around the time of the $\lambda$ wave (S4 Fig)—a potential considered the first visual evoked potential elicited after the suppression period occurring during the saccade [30]. Second, the accuracy of classifying the presence of motion–unlike the accuracy of classifying target motion direction–did not significantly vary across target motion directions ($F(4, 36)$ = 2.49, $\eta^2$ = 0.11, $p$ = .060, $p_{GG}$ = .091). This suggests that, although visual signals induced by intra-saccadic motion were present and could be decoded from EEG activity in all motion-direction conditions, their degree of usefulness for determining post-saccadic target location and executing timely secondary saccades varied across motion directions.

**2.2.4. Topography of decoding accuracy.** The third test was devised to investigate which electrodes most strongly drive whole-brain decoding of intra-saccadic motion direction. Assuming that motion streaks would be represented in the early visual architecture and inform secondary saccades in bottom-up manner, one would expect occipital electrodes to be the first not only to signal the post-saccadic target location, but also to encode a difference induced by continuous intra-saccadic motion. To investigate this hypothesis, we performed a time-resolved searchlight analysis, testing how well post-saccadic target locations could be decoded using just small subsets of electrodes. Indeed, a clear occipital topography arose between 50 and 80 ms after primary saccade offset (Fig 3a). After this initial visual response, starting around 110 ms after primary saccade offset, classifiers gradually started to exploit central and frontal sensors as well. Increased classification accuracy at frontal electrodes suggests that, at later stages, post-saccadic target locations could be decoded based on residual ocular activity introduced by corneoretinal dipoles when secondary saccades were made. In fact, if no correction for eye movement-related artifacts were performed (see Sect 5.7.2 for details), then classification of post-saccadic target locations becomes trivial due to the corneoretinal dipole signal: Classification accuracy for uncorrected data–after secondary saccades were initiated–approached 100%, regardless of target-motion direction (mean maximum classification accuracy; inward: $M$ = 92.7%, $SD$ = 3.4; outward: $M$ = 92.7%, $SD$ = 2.9; downward: $M$ = 89.0%, $SD$ = 3.1; upward: $M$ = 95.8%, $SD$ = 2.4). Prior to secondary saccades and with proper artifact rejection, a clear occipital topography was present, suggesting that the primary drivers of classification accuracy were visual signals, not eye movement-induced signals. As, arguably, an occipital topography cannot rule out the involvement of oculomotor activity in driving decoding accuracy, especially given the close temporal proximity between post-saccadic visual responses and the initiation of secondary saccades, we classified post-saccadic target locations and the presence of continuous intra-saccadic motion in a time window that strictly preceded the onset of each trial's secondary saccade. Despite temporal smearing introduced by the variance in secondary-saccade latencies, this additional control analysis revealed that both variables could be reliably decoded (even long) before secondary saccades were initiated (S12 Fig), further corroborating the notion that differences in visual activity, not in eye movements, allowed classifiers to differentiate between experimental conditions.

To confirm the role of visual signals, we examined the time courses of individual subsets of electrodes. First, as shown in Fig 3b, we found that occipital electrodes (defined as $X$ < −0.725 in cartesian location coordinates) yielded the highest mean classification accuracy ($M$ = 27.8%, $SD$ = 2.6), as compared to central (defined as −0.3<$X$ < 0.3; $M$ = 24.3%, $SD$ = 1.2) and frontal (defined as $X$ > 0.6; $M$ = 24.1%, $SD$ = 1.1) electrodes ($F(2, 18)$ = 35.2, $\eta^2$ = 0.50,

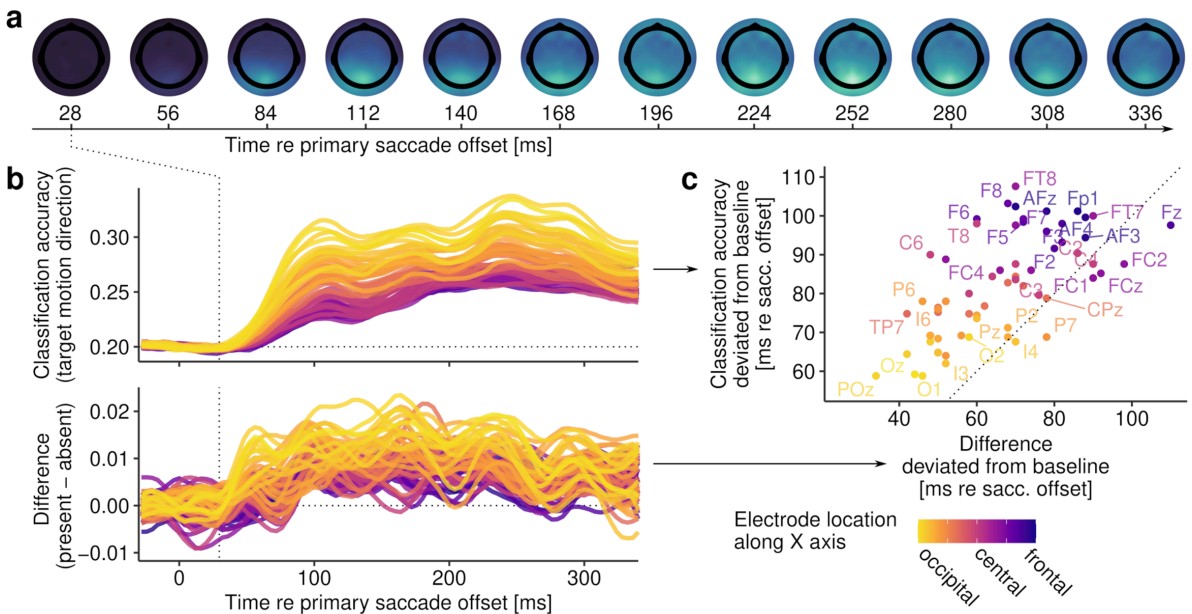

**Fig 3. Results of searchlight analysis. a** Topography maps show the spatial distribution of classification accuracy over time when determining target motion direction in systematically varied subsets of channels. **b** Accuracy of classifying target motion direction resolved for all channels, colored according to each channel's location along the cartesian X (FPz-Oz) axis. The upper row shows classification accuracy across motion present and absent conditions, whereas the lower row shows the respective differences in classification accuracy comparing present and absent conditions. **c** Scatterplot showing time points (extracted from the data in **b**) at which classification accuracy (y axis) or the difference in accuracy between conditions (x axis) first deviated from baseline activity, that is, when activity increased beyond the standard deviation of all channels' activity prior to primary saccade offset.

$p < .001$, $p_{GG} < .001$). Again, we found a clear main effect of continuous motion ($F(2, 18) = 74.9$, $\eta^2 = 0.04$, $p < .001$), but this benefit was significantly more pronounced for occipital than for central and frontal electrodes (occipital: $M = +1.1\%$, $SD = 0.2$; central: $M = +0.5\%$, $SD = 0.3$; frontal: $M = +0.4\%$, $SD = 0.3$; $F(2, 18) = 23.0$, $\eta^2 = 0.01$, $p < .001$, $p_{GG} < .001$). Second, occipital electrodes were also the first to capture differences between motion-present and motion-absent conditions after saccade offset. These differences, which were detected by an increase beyond one SD of the baseline differences across all electrode subsets of a given observer, could be decoded from occipital electrodes around 59.5 ms ($SD = 18.8$) after primary saccade offset, but only 79.2 ms ($SD = 30.9$) from central and 97.2 ms ($SD = 34.1$) from frontal electrodes ($F(2, 18) = 12.9$, $\eta^2 = 0.24$, $p < .001$, $p_{GG} < .001$). Finally, to determine whether this occipital-to-frontal progression holds across the entire population of sensors, we examined the relationship between the onset of successfully decoding a post-saccadic target location and the time when a difference between motion-present and motion-absent conditions first emerged (Fig 3c). Linear mixed-effects models revealed that there was nearly a unity relationship between these two temporal onsets ($\beta = 0.95$, $t = 2.97$, 95% CI [0.31, 1.59], $p = .026$), suggesting that the same signals that encoded post-saccadic target location also encoded the differential effect of continuous target motion. Evidently, these signals were primarily of occipital, thus early visual origin.

## 2.3. Coding of intra-saccadic target motion as orientation

EEG analyses have shown immediate post-saccadic visual signatures of intra-saccadic motion streaks which underpinned the facilitation of secondary saccades previous behavioral analyses

had revealed. Due to the low spatial resolution of EEG, one question has inevitably remained unanswered and can at this point only be addressed on the computational level: What is the nature of visual motion-streak signals? To resolve the dynamics of individual receptive fields (RFs), we created a large-scale filter model capable of simulating the spatial and temporal dynamics of early visual processing that provides–like a microscope–insights in the direct visual correlates of motion streaks. The model (illustrated in Fig 4a and described in detail in Sect 5.9) takes as input the exact stimulus and retinal trajectory and applies two banks of quadrature-paired log-Gabor filters–defined in terms of spatial frequency (SF) and orientation bandwidth–to compute spatial responses [31]. As a second step, SF-dependent impulse response functions limit spatial responses to a temporal-frequency (TF) range resolvable by the human visual system [32]. Third, as an energy mechanism, paired spatiotemporal responses are squared and subsequently undergo delayed normalization to accurately reproduce the temporal dynamics of transient and sustained responses in visual cortex [33]. The model's output thus equals the magnitude of visual responses resolved in (retinotopic) space and time, as well as across a range of SF and orientation channels.

By simulating visual responses to retinal target-stimulus trajectories in randomly sampled trials, we first used the model to understand the SF-orientation spectrum that is produced by the visual stimulation we applied experimentally. Fig 4b shows the average visual responses (collapsed across individual SF and orientation channels and over time) produced by each of the five target movement directions in a retinotopic reference frame. The panels show clear traces of activity that delineate the trajectory of the target, providing evidence that simulated RFs could well resolve the presence of the target, despite its high retinal velocities. Which channels predominantly encoded these traces? To answer this question, we aggregated responses over space and time, thus determining which channels were activated most strongly. Despite an overall large degree of similarity between the patterns (Fig 4c), there were systematic differences across target motion directions. Notably, the strongest overall activity was induced by outward target motion, which is not necessarily surprising: As the target moved in the same direction as the saccade, its retinal speed was heavily reduced (average minimum speed of only 29.8 dva/s, SD = 30.2, compared to 252.0 dva/s, SD = 70.7, in the static condition), leading to shortened streaks and longer-lasting exposure to fewer RFs. To determine the specific contribution of motion streaks to these patterns, we next compared motion-present with motion-absent conditions: The inspection of Fig 4d reveals that the SF channels most involved in motion streaks were in the mid-low range, that is, the SF range of the bandpass-filtered target noise patch (see Sect 5.4). More importantly, the orientation channels that predominantly encoded this contribution were strongly determined by the direction of the retinal trajectory. As indicated by the red crosses in the respective panels, RFs incidentally oriented in parallel to the target's motion direction almost entirely drove the representation of motion streaks. Only in the 'outward' condition this clear tuning was strongly reduced, again likely due to the low retinal velocities of the target (see above). These modeling results provide clear evidence that targets moving at saccadic velocities are encoded within a limited range of orientations parallel to the target's motion trajectory.

Within the framework of the proposed early-vision model, this predominance of parallel orientations evolves as a consequence of the limited TF range that the visual system is capable of resolving. To elucidate, we simulated the effect of a small stimulus (or even a single pixel) moving through a RF, oriented either orthogonal or parallel to the stimulus motion direction (Fig 5a). The stimulus could move at a low (10 dva/s) or high (100 dva/s) constant speed, producing patterns of responses as it passes through the RF (Fig 5b, 5c and 5d). At low speeds, both RFs were activated to a similar degree (Fig 5d), however, based on different filter responses (Fig 5b). In orthogonal units, the target moved through opposite polarities of the

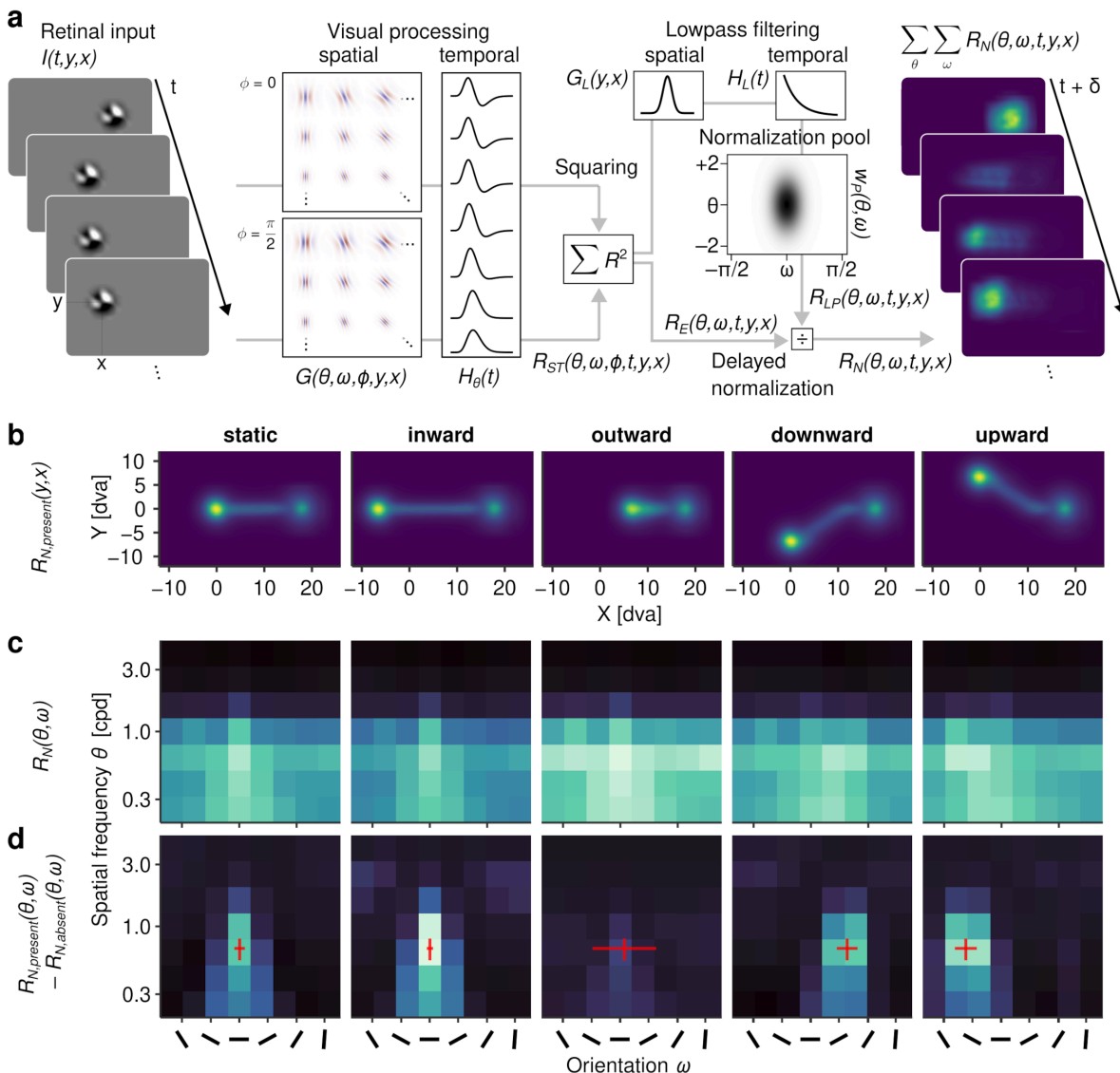

**Fig 4. Modeling early visual processing. a** Overview of the model setup. Retinal input was convolved with a bank of physiologically plausible spatial and temporal response functions (see Sect 5.9 for derivation), squared to achieve an energy output, and normalized using a variant of the delayed normalization model, yielding responses to retinal input resolved across space, time, as well as SF and orientation channels. **b** Spatial aggregates of model output (a.u.), collapsed over channels and time, showing the trajectory of the target in retinotopic space. **c** Engagement of SF and orientation channels (a.u.), summed over space and time and irrespective of the presence of continuous target motion. **d** Difference in engagement of SF and orientation channels between motion-present and motion-absent conditions. Red crosses indicate the average direction of the target's retinal trajectory and SF with maximum power, with error bars indicating $\pm 2SD$.

Gabor kernel–inducing TFs around 6 Hz, nearly ideal for the perception of motion [32,34]. In parallel units, the response was generated from the continuous presence of the target in one polarity of the RF, a mechanism also invoked by motion-streak detectors [35]. At increasing speeds, this inherent difference showed drastically: While in parallel units the response was reduced but remained clearly discernible as a brief burst, it was virtually absent in orthogonal units, rendering the target's fast motion unresolvable with this type of detector. This explains why responses in units oriented orthogonal to the retinal motion trajectory were so heavily

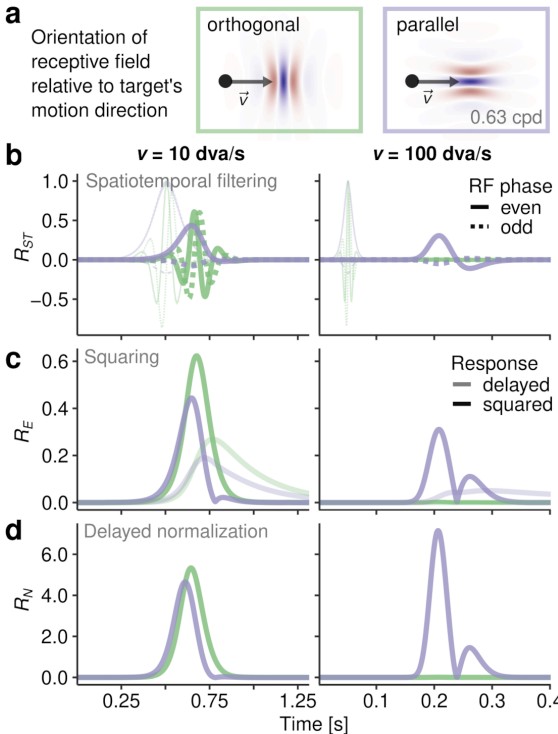

**Fig 5. Illustration of spatiotemporal processing of slow and fast motion. a** A target stimulus moves through two types of log-Gabor RFs, oriented either orthogonal or parallel to its direction of motion, at two different velocities. **b** Model responses over time after spatiotemporal filtering, computed for even (solid) and odd phases (dashed lines) of the Gabor filters. Transparent lines indicate the spatial response of the filter, prior to temporal filtering. **c** Squared responses and their delayed versions after temporal low-pass filtering (transparent lines), as used for normalization. **d** Responses after delayed normalization, the model output.

reduced (Fig 4d). This pattern arose merely as a consequence of the spatiotemporal characteristics of the visual system derived from psychophysical measurements during retinal stabilization (Fig 8). At the same time, the property of orientation-selective RFs to respond to motion parallel to their preferred orientation (see [36,37]) provides a parsimonious mechanism for the spatial coding of motion that would otherwise be too fast to be resolved.

If motion streaks are really coded as an orientation, as computational modeling suggested, then we can test this hypothesis by asking: Could this orientation signal be directly decoded from EEG data? To answer this question, we performed a final, exploratory MVPA to determine either clockwise (CW) or counterclockwise (CCW) direction of the intra-saccadic retinal trajectory of the target. CW or CCW labels were created by unique combinations of directions of primary saccade and target movement, respectively (Fig 6a). As these are labels defined purely in retinotopic space, their classification should be free of effects of motion direction in the spatiotopic reference frame. These classifiers could distinguish the two classes significantly better than with scrambled labels (Fig 6b) starting as early as around 32 ms when continuous motion was present, compared to 41 ms when it was absent (cluster-based permutation tests with $p < .001$). Moreover, classifiers performed significantly better when continuous motion was present (0–200 ms after saccade offset: $F(1, 9) = 18.0$, $\eta^2 = 0.08$, $p = .002$), first at 40 ms after saccade offset (cluster-based permutation tests; $p = .046$) and then with a prominent peak in classification accuracy at around 100 ms after saccade offset ($p < .001$).

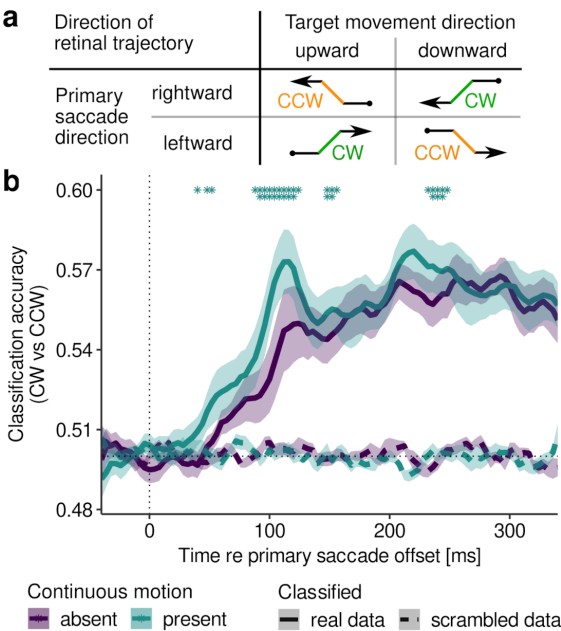

**Fig 6. Decoding retinal movement direction. a** Creation of labels CW and CCW to classify the direction of retinal trajectories during intra-saccadic target movement. **b** Classification accuracy as function of time relative to saccade offset when decoding CW vs CCW retinal directions in the presence and absence of continuous motion, respectively. Conventions same as Fig 2.

These results suggest that MVPA could well have picked up on an orientation-related signal–that would strictly be present during continuous motion–improving classification accuracy beyond what was possible by coding only motion direction.

## 2.4. Model-estimated visual dynamics predict secondary saccade latencies

While computational results improved our understanding how motion streaks are represented in the brain, we have not yet attempted to explain how they could be used to facilitate gaze correction. Previous proposals stressed the importance of object correspondence established via spatiotemporal continuity (cf. [16]), but whether this correspondence is necessary to explain the reduction of secondary saccade latencies is still unclear. As the model of early visual processing outlined previously allowed for a physiologically plausible quantification of visual responses in space and time, its estimates could now be used to test theoretical assumptions about the visual system's use of intra-saccadic information to speed up secondary saccades. Specifically, as not only absolute secondary saccade latencies, but also the effect of motion streaks on secondary saccade latencies varied considerably across target motion directions (cf. Fig 1c), it may well be possible that these differences could be explained in terms of early visual dynamics. To that end, we devised a computational model–coined the predictive switch model–to describe how intra-saccadic visual input may be used to infer the target stimulus' changing position across saccades (Fig 7a). The switch model assumes a system in which the current target position estimate $x_k$ is updated using (either predicted or measured) visual onset transients, that is, positive changes in the activity of spatial units (collapsed across SF and orientation channel for the sake of simplicity; for details see Sect 5.9.4). The system is capable of a time-resolved prediction of the visual consequences of its own saccade $\Delta \bar{x}_k^+$, as

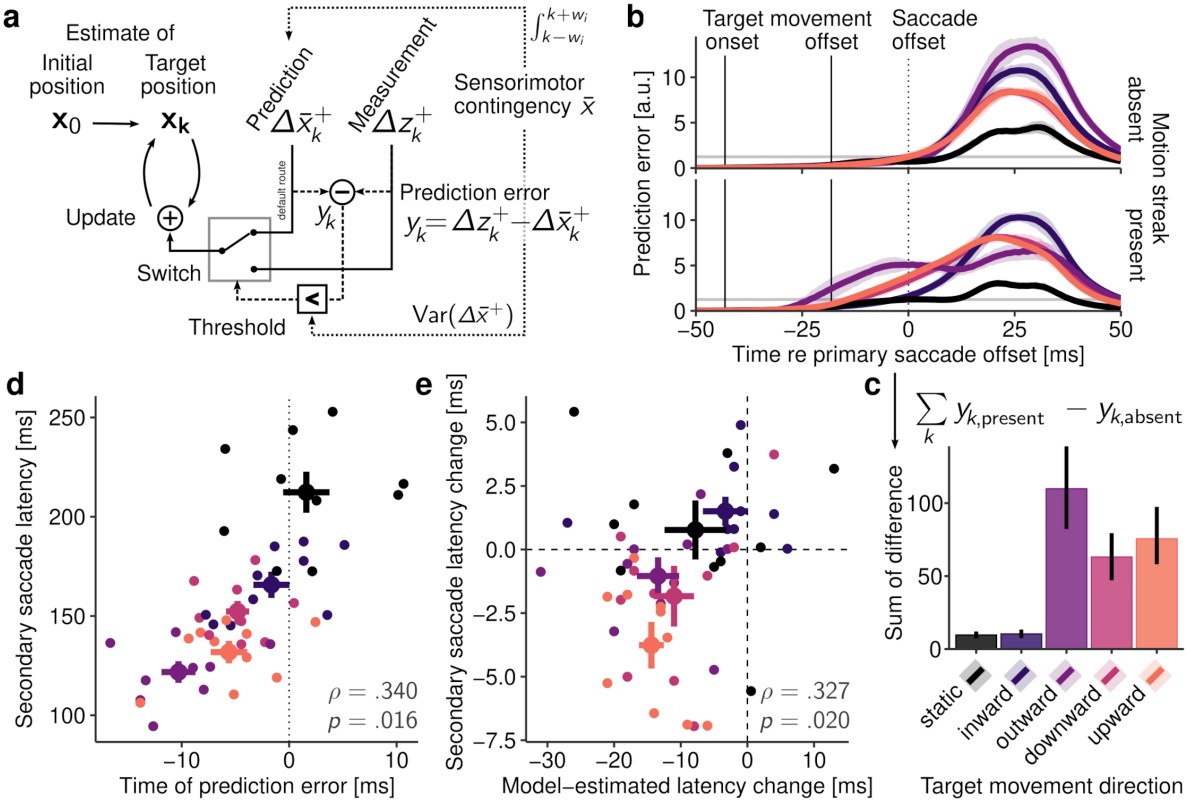

**Fig 7. The switch model and its predictions. a** Structure of the switch model. The target's position is iteratively updated using either predicted or measured visual changes. Predictions are generated from a learned sensorimotor contingency that also determines the threshold for switching between predictions and measurements, which occurs when an above-threshold prediction error occurs (see Sect 5.9.4 for details). **b** Prediction error as a function of time, aggregated across all participants, separately for all target motion directions. Transparent horizontal lines represent the average threshold for switching. **c** Positive sum of differences between present and absent continuous motion conditions. **d** Relationship between model-estimated time of prediction error and average secondary saccade latency measured in the corresponding experimental condition. Each data point represents a single participant, large dots show condition averages with $\pm 1SEM$, and Spearman correlation and significance level are printed in the lower right corner of the plot. **e** Relationship between differences in model-estimated times of prediction error (present–absent conditions) and corresponding differences between measured secondary saccade latencies (again present–absent conditions), plotted with the same conventions.

derived from the sensorimotor contingency $\bar{x}$. The latter was computed using the static condition, in which the target's retinal trajectory was exclusively determined by the saccade and not influenced by the experimentally injected target motion, thus representing an aggregate of experience with target-capturing saccades under normal viewing conditions (Fig 4b, leftmost panel). By default, the system can rely on its predictions to estimate the target's position, assuming a bias for stability [38,39]: Indeed, in most situations there is no need to actively localize and track the saccade-induced shift of an object, as an object can be assumed to not move considerably during the brief saccade interval, in which case the shift can safely be omitted from perception. At the same time, visual input around and during saccades is monitored by continuous measurements $\Delta z_k^+$ to detect a potential prediction error $y_k$ (Fig 7a). Large prediction errors that exceed a threshold of what could be expected from the typical variance of transients encoded in the sensorimotor contingency, the system 'switches' from relying on its internal predictions to measuring effective visual change (for details, see Sect 5.9.4). The time point at which this switch occurs essentially represents the moment when the system 'notices'

the displacement and starts taking into account sensory evidence in order to plan a secondary saccade to the displaced target.

Fig 7b displays the size of the prediction error for the major experimental conditions over time. Because, prior to experimentally induced target motion, activity patterns were highly similar across conditions, prediction errors were absent in the first half of the saccade and arose most prominently after the target shift, driven by the strong post-saccadic transients introduced once the target reached its final retinal position. Note that, due to variation in the saccade's landing position, above-threshold prediction errors also arose when targets remained static. Whereas prediction errors occurred largely at the same time in the absence of continuous target motion (Fig 7b, top), systematic differences emerged when continuous motion elicited motion streaks (Fig 7b, bottom): Outward movement elicited the earliest prediction error, again due to the increased activity in response to the target's reduced retinal velocity outlined previously, directly followed by upward and downward target motion. Inward target motion induced only late prediction errors. The benefit of intra-saccadic motion streaks thus again depended on the target's motion direction, with some resemblance to the secondary saccade latency results shown in Fig 1c. To determine whether this relationship holds across participants, we correlated the time of prediction error estimated by the model with the secondary saccade latency measured experimentally. Indeed, we found a significant positive correlation ($\rho = .340$, $p = .016$) between time of prediction error and absolute secondary saccade latency (Fig 7d). Furthermore, the degree to which continuous motion changed the onset of the prediction error predicted the corresponding change in secondary saccade latency ($\rho = .327$, $p = .020$; Fig 7e). These results not only speak in favor of the plausibility of the switch model, but also suggest a strong link between the experimental finding of motion-driven benefits in secondary saccade latency and the specific retinal dynamics that arise from combining saccade-induced and experimentally-induced target motion.

## 3. Discussion

Motion streaks result from objects shifting across the retina at high speeds. During saccades, motion streaks have been shown to improve the accuracy and reduce the latency of secondary saccades [16]. In this study, we set out to investigate the potential mechanisms underlying this effect. To this end, we combined eye tracking with EEG and created a physiologically plausible computational model to estimate the visual consequences of the experimentally induced retinal dynamics. Three major findings can be summarized.

First, on the behavioral level, we replicated the benefits for secondary saccade latency previously described [16]. That the benefit here was smaller may be attributed to the fact that only one target object was used (instead of six) and that no temporal constraints were placed on gaze correction by masking of object identities. We extended the finding by showing that it is specifically target motion orthogonal to the saccade's direction that results in a benefit. For target motion parallel to saccade direction, the case was more subtle. That is, for outward target motion, a small, but non-significant benefit was found. This may be related to the fact that secondary saccade latencies were the lowest in this condition, making a further reduction more difficult. In contrast, for inward target motion, a small, but significant cost of motion streaks was found. Finally, in the absence of target displacement, it did not matter whether continuous motion was present or not. Using computational modeling, we were able to explain these multilayered experimental results based on the neural dynamics resulting from motion streaks.

Second, on the neuronal level, we found that it was well possible to decode high-speed and strictly intra-saccadic target motion from the EEG signal, providing strong evidence

that visual input presented during saccades is not removed from visual processing. In the early post-saccadic interval, well before 100 ms, and regardless of the target's motion direction, the EEG signal contained information as to whether continuous, 'streaky' target motion or merely step-like target displacement was presented, even though perceptually these two types of motion are very hard to discriminate [14]. Importantly, even when classifiers were trained on step-like target displacements only, so that had never 'seen' any motion streaks, continuous motion still allowed for an earlier decoding of post-saccadic target location. This finding is remarkable because targets arrived at their final location at the same time regardless of motion type, which suggests that the moving target's position information was integrated throughout the saccade. How well this decoding was possible, however, depended on the target's motion direction: Directions orthogonal to the saccade yielded higher classification accuracy than parallel directions. On a qualitative level, this finding mirrors the direction-dependence of latency benefits described above and suggests a common origin.

Third, on the level of computational modeling, we explained our experimental findings by combining a model of early visual processing that was exclusively informed by neurophysiological and psychophysical studies with the switch model, a more general framework to understand how the visual system may treat this processed information, to predict secondary saccade latency. Addressing how motion streaks can be coded, the early visual processing model highlighted the relevance of orientation-selective channels in the representation of motion streaks and showed that responses of these RFs allow a spatial coding of the target's motion trajectory even at saccadic velocities and beyond. Addressing how motion streaks may be used to facilitate gaze correction, the switch model took as input the visually processed trajectory of the target in retinotopic coordinates and estimated intra-saccadic prediction errors. The temporal onsets of these errors matched the experimental patterns of secondary saccade latencies surprisingly well. This suggests that, in order to perform timely gaze correction, it is crucial to notice prediction errors as early as possible by comparing learned saccade-induced patterns (with static targets) with concurrently monitored visual activity. In this sense, latency differences across experimental conditions stem from different timings of significant prediction errors. Although we cannot rule out additional influences of potential asymmetries in the motor domain, our modeling suggests that these differences between conditions may be fully reducible to the differences in visual patterns that arise in each of them: The more compatible a measured pattern is with the learned representation of a saccade-induced visual consequence, the longer it takes for the system to realize that a significant deviation from its prediction has occurred, leading to a delayed accumulation of sensory evidence of the new target location and, ultimately, longer gaze-correction latencies.

## 3.1. Prediction of saccade-induced visual consequences

As a framework aimed at describing how the visual system actively monitors saccade-induced input, the switch model is based on three fundamental premises, each supported by existing experimental evidence. First, and obviously, we assume that visual input during saccades is not discarded from processing, but actively monitored to create a representation of the sensory consequences generated by the executed eye movement–the sensorimotor contingency. Indeed, there is mounting evidence not only that intra-saccadic stimulation has immediate motor and perceptual consequences (for a review, see [40]), but also that habituation to stimulation during saccades has an impact on peri-saccadic sensitivity to detecting flashes [41], large-field displacements [42] and motion [43]. Second, based on these previous findings, it is also reasonable to assume that sensorimotor contingencies can be used to generate predictions about the sensory outcome of a saccade, which give rise to prediction

errors, for instance, if saccade targets are displaced during eye movements [44–46]. Prediction errors have been suggested to be useful to detect visual target offsets upon saccade landing and adjust motor commands [47] or even update perceptual thresholds [42], and here we merely extend this principle to the intra-saccadic interval. Third, intra-saccadic visual transients, which occur routinely during natural vision [48], are omitted from perception as long as they are compatible with the sensorimotor predictions, and are readily rendered visible once a significant prediction violation occurs. Strong support for this assumption comes from the finding that high-intensity transients during saccades (applied in virtually all experimental paradigms) can faithfully be reported (e.g., [7,49]) and, even though with systematic errors, localized in head-centered space (e.g., [50–52]). This is contrasted by the equally ubiquitous observation that visual consequences of saccades, when no externally introduced transient occurs, are omitted from perception. Indeed, static and continuous targets with peri-saccadic offsets are not (mis-)localized, suggesting that there is neither a need for the maintained visual localization of static targets, nor for localization of offset transients [53]. This is also the reason why the switch model considers only onset transients. Combining these three premises, the proposed switch model makes an interesting preliminary case how the seeming contradiction of the omission of saccadic visual consequences and the exploitation of intra-saccadic visual signals may be realized: Visual transients are omitted as long as they are compatible with the sensorimotor contingency, but may be readily rendered visible and localized once a (significant) deviation from it occurs.

It is a long-standing question why the transients induced by the shift of the entire visual image during saccades do not disturb stable and continuous vision. While active accounts of omission invoke that transients remain omitted from perception as long as they remain predictable, passive accounts of omission rely on visual-only contributions, such as smearing and masking, to reduce the relevance of these transients [8,9]. Our framework suggests that these accounts are not mutually exclusive. On the one hand, if a sensorimotor contingency is learned from the visual consequences of one's own eye movements, then this learning would also entail the ubiquitous effect that the static pre- and post-saccadic scenes should have on the visual signal. To effectively control for the largest saccade-induced visual transient, which is produced by the onset of the static post-saccadic image, the sensorimotor contingency must also contain it. Very similar to the sensorimotor contingency modeled here, Teichert and colleagues [54] also created a representation of the saccade-induced visual consequences by taking into account the entire continuous shift of visual targets, including their static pre- and post-saccadic locations. The importance of the latter is evidenced by the finding that pre- and post-movement end points were capable of rendering perceptual thresholds for detecting high-speed object movement amplitude-dependent [29]. On the other hand, active processes of omission may well be facilitated by pre- and post-saccadic masking, if the latter is understood as a mechanism that reduces the strength of intra-saccadic transients, similar to the impairment of peri-saccadic contrast sensitivity known as saccadic suppression [13,55,56]. That is, dampening intra-saccadic vision could reduce the probability of accidental prediction errors that may arise due to spurious above-threshold mismatches between predicted and measured visual patterns. Specifically during natural vision, due to full-field visual scenes containing high power across SFs and orientation, such dampening would be expected to be considerably stronger than in the experiment presented here, which used only a single target on a uniform background. Therefore, instead of viewing active and passive mechanisms of saccadic omission as antagonists, a framework based on sensorimotor contingencies allows a new line of investigation into how they could align.

Finally, given the imperfection of the oculomotor system, the potentially unlimited space of observable stimulus patterns, and the complex spatiotemporal transformations they

undergo around eye movements, how could the system possibly predict an impending saccade's sensory consequences in an accurate manner? The switch model yields a surprising tentative insight into this question. In initial simulation attempts, we found that if visual activity in a given trial was compared to the sensorimotor contingency in perfect temporal alignment, prediction errors were extremely frequent, rendering any omission based on this comparison highly ineffective. These prediction errors clearly arise from various sources of noise in the system, related to the variance in saccade trajectories, stimulus features and timings, as well as early-visual neuronal response patterns (e.g., [57,58]). A counter-intuitive solution to this was to integrate over a larger temporal window when extracting the prediction from the sensorimotor contingency: By allowing greater temporal uncertainty, and thereby also greater spatial range, only those mismatches that arose from incompatibilities between predictions and measurements signaled a prediction error, increasing the reliability of omission. Even though at this point we can only speculate about the precise nature of predictions derived from sensorimotor contingencies, this theoretical consideration suggests that a coarse and stereotypical prediction may be a more parsimonious and computationally inexpensive solution to active omission than a precise and reafference-like prediction (for a more detailed discussion, see [8]).

## 3.2. Relevance of intra-saccadic motion streaks

Experimental eye-tracking and EEG findings have provided insights into the benefits of continuous target motion that was (unlike step-like target motion) capable of inducing intra-saccadic motion-streak patterns. Capitalizing on the high temporal resolution of EEG to investigate the temporal dynamics of cortical activity that arose from such streak patterns, our study critically depended on decoding analyses. MVPA has successfully been applied to the investigation of saccadic remapping, that is, the process by which neurons predictively respond to stimuli that will be in their RF upon landing of the impending saccade [59]. For example, classifiers were trained on neural responses to different stimulus categories (such as faces vs houses or gratings with CW vs CCW orientation) viewed during fixation and then tested at various time points around the saccade, revealing the time course of information transfer across saccades [60,61]. Other studies tested the spatial specificity of remapping, training classifiers to detect the presence of a stimulus at different screen locations, and then determining when before a saccade remapped activity became available if the location corresponded to the stimulus' future (post-saccadic) retinotopic location [62]. Due to the strictly intra-saccadic stimulus presentations we applied, our study cannot make a statement about remapping. Yet, our experimental approach pursued a similar goal, that is, to determine at what time and processing stage critical neural signatures–in our case, of motion streaks–first emerge. Importantly, even though motion streaks during saccades were task-irrelevant and perceptually hardly noticeable [14], decoding of not only their presence, but also their orientation was possible within less than 100 ms after saccade offset in every motion-direction condition. Corroborated by their occipital topography, these results suggest that intra-saccadic visual signals are processed together with the visual image of the new fixation– and from there could facilitate the programming of visually guided secondary saccades. Even though behavioral benefits of motion streaks have been reported previously [16], a mechanistic explanation was still missing. Our computational modeling attempted to close this gap. The physiologically plausible early-visual processing model provided strong evidence that spatial location of the target's motion trajectory over time is encoded in channels tuned to orientations parallel to the target's motion direction. Due to the fact that RFs were only stimulated very briefly, the evoked visual activity was weak when compared to the activity evoked by largely stable

stimulation at the end of the saccade. Yet, spatial traces were clearly discernible by model output and their orientations could even be decoded from the EEG signal. The notion that fast motion can be coded in orientation-selective channels is well supported by previous results from psychophysics [35,63,64], imaging [65], and neurophysiology [36,37]. For the special case of intra-saccadic motion streaks, such a mechanism would explain why perceptual sensitivity to continuous (as opposed to step-like) motion depended so strongly on the target's relative orientation [14] or why specifically targets with high power in orientations parallel to their retinal trajectories induced benefits for gaze correction [16].

This visual mechanism of coding motion as orientation provokes an interesting thought: When learning the visual consequences of its own saccades, the system could employ the rule that each saccade made across a largely static visual scene essentially bandpass-filters the scene's content according to that saccade's direction and velocity [8]. Indeed, our visual processing model indicated a clear activity peak for horizontally oriented RFs when horizontal saccades were made to a static target. When expecting activity in horizontal-selective channels, observing peak activity in CW- or CCW-selective channels instead should produce a strong prediction error. This may well explain why both location decoding from the EEG signal and enhancement of gaze correction was most efficient for target motion orthogonal to saccade direction: By the engagement of orientation channels that would otherwise not be stimulated by a horizontal saccade, activity could only have been induced by stimulus motion in the external world, not by the saccade. The same argument would be valid for the case of inward target motion, for which continuous motion produced a negative benefit for gaze correction and no benefit in the decoding of post-saccadic target location: Modeling showed that the trajectory of inward target motion would not only be very compatible with the trajectory of a static target (especially in the case of an overshooting saccade), but also engage the same orientation channels as a static target, rendering it nearly impossible to detect a displacement prior to the end of the saccade. This example suggests that the system may also rely on rather coarse representations about the consequences of its own saccades to compute a prediction error.

While the visual dynamics produced by combining saccadic and target motion seem to play an important role, the oculomotor system may also contribute to aggravating these tendencies. It is a ubiquitous finding that saccadic landing positions exhibit larger variance along the radial than the tangential axis–an anisotropy that is also reflected by sensitivity to intra-saccadic displacements [19], even across individuals [66]. If such information is also coded in the sensorimotor contingency, then derived predictions should show larger spatial uncertainty around the end of the saccade, making the detection of radial (in our case horizontal) target offsets more difficult than tangential (vertical) offsets. This principle directly generalizes to the intra-saccadic interval because the saccade's trajectory and velocity profile is scaled to match its resulting amplitude [67], thus again making parallel target motion less salient than orthogonal target motion. An exception to this may be outward target motion, if it can–like in our paradigm–greatly reduce the target's retinal speed. In this case, activity evoked by the target increases greatly and a prediction error occurs as a consequence. In the special case of brief retinal stabilization, the target can appear like a brief intra-saccadic flash. Some of our participants indeed reported experiencing this. In addition to an early prediction error to trigger target localization, outward target motion also simulates the most common case, that is, saccadic undershoot [68,69]. It is thus well possible that extraretinal errors, too, contribute to the generation of low-latency secondary saccades in this condition. Finally, it may present a curious case why upward secondary saccades are generated slightly faster than their downward counterparts–a finding that the our modeling also reproduced. The model picked up on a tendency present in primary saccades, that is, a small upward component in horizontal

saccades, amounting to 0.15 dva on average (S8 Fig), which caused targets to travel slightly into lower visual field. Being a mere consequence of the saccade program, this tendency was encoded in the sensorimotor contingency, rendering prediction errors due to downward target motion slightly less salient to detect. That said, our model underestimated the actual latency difference between upward and downward secondary saccades, as it did not take into account possible benefits induced by the retinal eccentricity of the target. That is, the closer primary saccades landed to their (displaced) post-saccadic target, the faster secondary saccades could be initiated [16]. In the present study, primary saccades landed slightly above the pre-saccadic target location, consequently producing a benefit for upward and a cost for downward secondary saccades. Taken together, even though our modeling input included no more than the visual activity arising due to the retinal trajectory of the target stimulus, the predictions of the switch model showed surprisingly strong correlations to our empirical results.

In our modeling framework, the role of continuous, 'streaky' target motion can be boiled down to a simple rule: The onset transients introduced by it lead to an earlier detection of external stimulus motion. At what point in time the pattern of these transients diverges from the sensorimotor prediction, however, depends strongly on the resulting retinal trajectory of the target. Note that, also for reasons of parsimony, we have not modeled the processes that occur from the onset of the visual prediction error to the execution of the secondary saccade. Clearly, earlier prediction errors allow for a prolonged evidence accumulation time when estimating post-saccadic target location, but it remains an open question whether there is a specific benefit of continuous motion, for instance, because it resembles the visual input during natural vision the most. However, there is no reason to assume that an earlier target displacement (i.e., without a blank to match continuous motion duration) could not already suffice to produce a similar or greater latency benefit, making this question an interesting subject for future research.

## 4. Conclusion

What is the role of saccade-induced visual input and how could the visual system make use of it? We have approached this question by investigating potential benefits of continuous motion during saccades, as such spatiotemporal continuity should occur as a natural consequence of saccadic eye movements, and because it has been claimed that the visual system should be optimized for the localization of continuous, as opposed to flashed stimuli (e.g., [54]). Our EEG results suggest that the benefit observed for gaze correction is likely related to the early-visual benefit in the enhanced representation of the post-saccadic target location and target motion direction, as these signals consistently preceded the execution of secondary saccades and were present predominantly in occipital sensor locations. Our modeling results further suggest that this early-visual benefit could be grounded in coding fast motion in orientation channels, thereby creating a spatial code for the trajectories of objects that change their retinotopic locations across saccades. Finally, by comparing the visual patterns evoked by these trajectories to an aggregated representation of saccade-induced visual consequences, the system becomes capable of distinguishing self-induced and externally-induced target motion, one of the major challenges in visual stability [70,71]. This challenge is, at least on the phenomenological level, tightly linked to the challenge of omitting self-induced visual changes. Our work may not answer the question whether it is specifically continuous motion, as opposed to step-like motion, that drives these benefits. Yet, using the example of motion streaks as a ubiquitous intra-saccadic visual pattern, it provides a tentative perspective–along with experimental support–on how the visual system may represent and utilize these signals

while, at the same time, omitting them from perception. This perspective may eventually inspire future research on the intriguing questions of saccadic omission and visual stability.

# 5. Materials and methods

## 5.1. Ethics statement

The experiment was conducted in agreement with the latest version of the Declaration of Helsinki (2013) and was approved by the Ethics board of the Department of Psychology at Humboldt-Universität zu Berlin (protocol number 2018-36, "Mechanismen visueller Stabilität"). All participants provided written informed consent prior to their participation in the study.

## 5.2. Participants

Ten observers (five female, age range [21, 33]), one of them being the study's first author, were tested in the EEG-eye tracking experiment. All had normal or corrected-to-normal vision (i.e., 20/20 ft. acuity in the Snellen test), four wore glasses, and one person wore contact lenses. Five had right ocular dominance, which was tested with a variant of the Porta test, and all were right-handed. The experiment was pre-registered at OSF (https://osf.io/n5qrm).

## 5.3. Apparatus

The experiment was performed in a dimly lit, sound-attenuated, and electromagnetically shielded cabin, using a PROPixx DLP Projector (Vpixx Technologies, Saint-Bruno, QC, Canada) with a temporal resolution of 1440 frames per second and a spatial resolution of 960 x 540 pixels that projected into the cabin onto a 200 x 113 cm screen (Celexon HomeCinema, Tharston, Norwich, UK). The screen covered a space of approximately 63.5 x 35.8 degrees of visual angle (dva) as observers were seated at a distance of 180 cm, their heads supported by a chin rest. The stimulus display was realized using PsychToolbox [72,73] in Matlab 2016b (Mathworks, Natick, MA, USA) on a custom-build desktop computer with an Intel i7-2700K eight-core processor and a Nvidia GTX 1070 Ti graphics card, running Ubuntu 18.04.1 (64-bit) as operating system. Eye tracking was performed using an Eyelink 1000+ desktop base system, tracking participants' dominant eye at a sampling rate of 2000 Hz with the help of the Eyelink Toolbox [74]. EEG data was collected with a total of 70 electrodes (64 EEG using a custom cap, 2 Mastoid, 4 EOG, see https://osf.io/jasrk for all channels) using an actiChamp EEG amplifier (Brain Products, Gilching, Germany) at a sampling rate of 2500 Hz. For data collection, the left mastoid electrode was used as a reference electrode. To synchronize eye tracking and EEG data, we applied a DB-25 Y-splitter cable to simultaneously send triggers of 1.1 ms duration to both Eyelink and EEG host computers. To temporally align EEG and Eyelink recordings (see Sect 5.7.2), we used the EYE-EEG Toolbox, version 0.85 [20] in EEGLAB, version 14.1.1 [75].

## 5.4. Stimuli

Target stimuli were noise patches bandpass-filtered to a mid-low spatial frequency (SF) range with -3dB cutoffs at 0.33 (SD = 0.10) and 1.02 (SD = 0.12) cycles per degree of visual angle (cpd). They were of 100% Michelson contrast and were enveloped in a Gaussian aperture with a standard deviation of 0.56 dva. To facilitate decoding, the same noise patch was used for each participant in all trials. Fixation markers were white squares of 0.44 dva width and height, containing white centers. All stimuli were presented were presented on a uniform

grey background with a luminance of 30 $cd/m^2$. Intra-saccadic motion of the target stimulus had a duration of 25 ms (36 frames at 1440 fps) and covered a distance of 6.6 dva at constant velocity.

## 5.5. Procedure

Trials started with the presentation of the fixation marker on either the left or right side of the screen and the target stimulus on the opposite side of the screen at an eccentricity of 18.25 dva. Fixation checks were performed for 400 ms within a circular boundary with a radius of 2.2 dva around the fixation marker. Trials were aborted and repeated at the end of each run after 2 seconds of unsuccessful fixation or more than 100 refixations. Upon successful fixation, the fixation marker disappeared, which for observers constituted the cue to make a saccade to the target stimulus. As soon as the saccade was detected online (using the algorithm described in [76], with parameters $\lambda = 8$, $k = 3$, $\theta = 30$), the target could be either displaced to four different locations (upward, downward, outward, and inward conditions), each at 6.6 dva distance relative to the initial target location, or remain at its initial location (static condition). This displacement was either be a linear, continuous movement of the target (streak present condition) or a blank screen for 25 ms duration (streak absent condition). After the primary saccade was concluded, participants made secondary saccades to the displaced target stimulus. Other than that, no response was required, and trials were terminated automatically 450 ms after the target reached its final location. Feedback was provided if no primary saccade was made within 10 seconds after cue onset ('You have not made a saccade!'), if a secondary saccade did not reach a 3.3-dva boundary around the final target stimulus location within 450 ms after stimulus movement offset ('You have not reached the final stimulus location!'), if gaze position did not fall within a 2.8 dva circular boundary around the saccade target ('You have not reached the final stimulus location!'), or if two or more saccades were made instead of one primary saccade ('Please make one saccade (you made XX)'). Trials, in which any of these events were detected, were repeated at the end of each run.

## 5.6. Experimental design

A single experimental session was subdivided into four runs, each containing two blocks of 250 trials. Each of the ten participants performed 2 sessions, that is, 8 runs of 500 trials each, resulting in a total of at least 4000 trials. We tested three orthogonal experimental conditions in a 5×2×2 design, resulting in 200 trials per experimental cell, a trial number well capable of achieving decent statistical power in ERP experiments, even when testing only ten observers [77]. First, movement direction (5 levels) determined whether target stimuli were displaced upwards, downwards, outward (i.e., in saccade direction), inward (i.e., against the saccade direction), or remain static. Second, movement was either continuous (i.e., moving at constant velocity, thus producing a streaky intra-saccadic trajectory) or apparent (i.e., blanked and re-appearing at target location after 25 ms). Third, saccades (18.25 dva instructed horizontal amplitude) were either of rightward or leftward direction. All trials were presented in a randomly interleaved fashion, but rightward and leftward saccades were made in alternating order to reduce time-consuming and costly large gaze shifts between trials.

## 5.7. Preprocessing

**5.7.1. Preprocessing of eye-tracking data.** As a first step, all those trials were excluded in which fixation control was not passed or presentation frames were dropped during target motion. Such trials were rare, leading to an a priori exclusion of 0.19% (range [0.09, 0.40]) of

all trials. As a second step, saccade detection was performed individually for each trial, taking into account all samples for cue onset to the end of the trial. Saccades candidates were detected on eye movement data sampled at 1000 Hz using the Engbert-Kliegl algorithm [78, 79] with $\lambda = 6$ and a minimum saccade duration of 5 samples. To avoid erroneous detection of secondary saccades, clusters of detected saccades separated by less than 30 samples were merged to one saccade. To achieve a conservative estimate for saccade offset, post-saccadic oscillations (PSOs)–artifacts induced by inertial forces acting on the iris and lens [80]–were detected using a procedure described in [81], in which saccades are parsed for PSO onset, using both direction-inversion ($\theta = 90$ deg) and minimum-velocity criteria. Owing to the fact that PSOs have considerable durations and can therefore greatly distort estimates of saccade duration and the empirical finding that PSO peaks constitute more reliable estimates for the physical end of the eyeball's rotation [81,82], we defined saccade offset as PSO onset (preferring onsets detected by direction inversion over those detected by minimum velocity due to their higher fidelity), if a PSO was detected. If a PSO was not detected, which in fact occurred in the majority of trials (M = 65.5%, range [8.5, 89.1]) due to instantaneous velocity falling below threshold around PSO onset or PSOs not being detected as saccade candidates in the first place, generic saccade offsets were used. For trials to be included for analysis, they had to conform with three major criteria. First, primary saccades had to be initiated after cue onset, land in the vicinity to the initial target location (as defined by a circular boundary with a radius of 3 dva around the target), not contain missing eye tracking samples, and have reasonable metrics (i.e., durations between 30 and 100 ms and peak velocities between 100 and 1000 deg/s). Second, motion of the target stimulus had to be initiated after saccade onset and concluded prior to saccade offset (each time taking into account a deterministic video delay of the projector of one refresh cycle; [76]), rendering target motion strictly intra-saccadic. Third, secondary saccades had to be made and land in the vicinity of the final target location (again defined by a 3-dva circular boundary), not contain missing samples, and yield peak velocities between 100 and 1000 deg/s. Following this procedure, another 10.9% (range [3.1, 25.1]) of all trials had to be excluded. Valid tertiary saccades occurred in only 17.8% (range [4.0, 36.6]) of all trials with secondary saccades and were therefore not further analyzed.

**5.7.2. Preprocessing of EEG data.** EEG data was downsampled to 1000 Hz, filtered by a Parks-McClellan notch filter around 50 Hz to remove line noise and a second-order Butterworth bandpass filter from 0.016 to 100 Hz, and finally re-referenced to an average reference. Note that the high-pass frequency of 0.016 Hz was set low enough to not gravely distort the time series of decoding analyses [83]. Using the EYE-EEG Toolbox [20], we synchronized eye movement data with EEG data, thus downsampling eye movement data (recorded at 2000 Hz to minimize online saccade-detection latencies; [76]) to 1000 Hz to match sampling frequencies. EEG and eye-tracking samples could be matched with a high degree of accuracy (0.41 ms mean absolute error, range [0.36, 0.46], see S1 Figa) and cross-correlation analyses revealed that both recordings were indeed well aligned (0.4 ms mean maximum cross-correlation lag, range [−1, 3], see S1 Figb).

To remove ocular EEG artifacts, in particular corneoretinal dipole and spike potentials, from the data, we ran an optimized ICA procedure, themed OPTICAT [84], using a high-pass frequency of 3 Hz and an overweighting factor of 0.5 for spike potentials. ICA training was performed on epochs ranging from -0.6 to 0.5 seconds relative to each trial's target motion onset trigger. Resulting weights were applied to preprocessed data and ICA components were removed (only) based on the saccade/fixation variance-ratio criterion: If the variance ratio of saccade and fixation periods–the latter defined as starting at fixation onset until 10 samples prior saccade onset to compensate for potentially later saccade detection due to elevated

velocity thresholds–was larger than 1.1, the component was likely driven by eye movements and therefore rejected [85]. As shown in S2 Fig, this procedure achieved a complete removal of the effects of the corneoretinal dipole on the EEG signal, as well as a virtual absence of the spike potential prominent around saccade onset across all ten observers (cf. Fig 4 in [84]). Finally, after saccade detection was performed on eye tracking data (see Sect 5.7.1), combined EEG and eye-tracking data were downsampled to 250 Hz to reduce the computational complexity of subsequent decoding analyses.

## 5.8. Analyses

**5.8.1. Secondary saccades.** Statistical significance testing for primary and secondary saccade metrics and stimulus timing was performed using repeated-measures ANOVAs, as implemented in ez R package [86]. In case of a significant Mauchly's sphericity test, Greenhouse-Geisser corrected p-values ($p_{GG}$) are reported along with p-values. We computed within-subject standard errors of the mean (SEM) using the method of Cousineau [87]. To approximate ANOVA MSE, the correction by Morey [88] can be applied: In the our case of ten within-subject conditions, the correction would amount to a factor of 1.11.

**5.8.2. Decoding.** MVPA analyses were performed to classify three types of target classes: (1) post-saccadic target location (5 classes: static, inward, outward, downward, upward), (2) presence of continuous motion (2 classes: absent, present), and (3) retinal motion direction (2 classes: CCW, CW; computable in upward/downward conditions only). The features used for classification were the 64 EEG channels (without EOG and mastoid channels), which were baseline-corrected using a window of 40 to 5 ms before saccade offset. Classification was performed using L2-regularized logistic regression, as implemented in the LiblineaR package [89] with default parameters, separately for individual observers, primary saccade direction, and non-target experimental condition (that is, movement direction when decoding of continuous motion and presence of continuous motion when decoding target movement direction, respectively), and for each time point. Time was aligned to primary saccade offset. In order to achieve uniform sampling with respect to these events without needing to create time bins, EEG and eye movement data were spline-interpolated.

Classification was performed in a leave-one-out fashion, in other words, each data sample was tested individually by a model trained on all other data samples with the same time stamp. Control classifications, in which target classes were shuffled, were also performed to estimate the baseline variance in classification accuracy around chance level. At each iteration training data was centered and scaled, and it was made sure that training data contained an equal amount of samples in each class by removing randomly chosen samples from those classes with excess trials. For each sample the predicted class was saved, as well as the channel weights used to perform classification. To determine activation patterns for topography plots (created using R package eegUtils [90]), classifier weights were multiplied with the covariance matrix of the training data [91], following the implementation of the MNE Python library [92]. In the case of multinomial classification, LiblineaR uses the one-versus-rest approach (i.e., the model discriminates between one class and all other classes for each class), in which case we saved pattern estimates of each trial's target class. In searchlight analyses we performed the same procedure, but for each channel selected a subset of channels, that is, the current target channel plus its four closest neighbors which were selected based on their distance to the target channel in XYZ space (maximum allowed distance was set to 0.6).

After classification, classification accuracy (i.e., the proportion of correctly classified trials) was computed individually for each observer, experimental condition, and time point. To reduce sample-to-sample noise in classification accuracy, individual time courses were

smoothed with a five-point moving average. To test for differences between conditions, we applied mixed-effects cluster-based permutation tests [93], as implemented in the permutes R packages [94], allowing random effects by participants and running 2000 permutations. To analyze temporal onsets of significant decoding, we also applied linear mixed-effects models with random intercepts and slopes [95]. In these cases, parametric bootstrapping (5000 repetitions) and Satterthwaites's method of computing degrees of freedom and t-statistics were used for test for non-zero weights.

## 5.9. Modeling

To goal of the modeling was to simulate visual responses to practically any (monochromatic) spatiotemporal stimulus–in this case, the rapid movement of a noise patch across the retina–using physiologically plausible spatial and temporal response functions. To that end, we first created a matrix representation of the retinotopic stimulus positions over time $I(t,y,x)$ (see Fig 4a for an illustration): At each time point $t$, we inserted the spatial stimulus–represented as a 2D matrix with horizontal and vertical dimensions amounting to 51 pixels, which corresponds to 3.36 dva–by locating its center at the estimated retinotopic location. The latter is given by subtracting gaze position (resampled to 1440 Hz to match the projection system's temporal resolution) from the concurrent placement of the stimulus center.

**5.9.1. Spatial response functions.** To simulate responses to spatial input we created a bank of log Gabor filters $G(\theta, \omega, \phi)$ with seven spatial frequencies $\theta$

$$S_\theta = \{0.25, 0.4, 0.63, 1.0, 1.59, 2.52, 4.0\} \text{ cpd} \tag{1}$$

and eight orientations $\omega$

$$S_\omega = \{-67.5, -45, -22.5, 0, 22.5, 45, 67.5, 90\} \text{ deg} \tag{2}$$

as well as even and odd phases ($\phi$; levels: 0 and 90 deg), as Gabor filters have been shown to be very suitable models for early-visual receptive fields (RFs; [96–98]). Based on the procedure described in [31], log-Gabor filters were constructed in the frequency domain, where they correspond to a Gaussian profile with standard deviations $\sigma_\theta$ in log-frequency and $\sigma_\omega$ in orientation space. Schütt and Wichmann [31] set $\sigma_\theta$ to a constant value of 0.5945 octaves (∼0.7 octaves half width at half maximum, HWHM) and $\sigma_\omega$ to 0.2965 (∼20 deg HWHM), but neglected the fact that SF and orientation bandwidth increases at low spatial frequencies–otherwise, if kept constant, the visual system would have to implement extremely large RFs to accurately localize low-SF stimuli in frequency space.

To model the increase of SF bandwidth in the low-SF domain, we relied on the finding that bandwidth remains largely constant at SFs equal or larger than 2 cpd, while below this value bandwidth increases by 0.26 octaves for each octave of SF [99]. SF bandwidth $w_\theta$, that is, the full bandwidth at half maximum (FWHM), would thus depend on spatial frequency $\theta$ in the following manner:

$$w_\theta(\theta, \theta_0, w_0) = \begin{cases} w_0, & \text{if } \theta \geq \theta_0 \\ -0.26 \log_2(\theta/\theta_0) + w_0, & \text{if } \theta < \theta_0 \end{cases} \tag{3}$$

where $\theta_0$ is the critical SF (i.e., 2 cpd) and $w_0$ is the asymptotic FWHM, that is, 1.5 octaves, as estimated by Anderson and Burr [99], which corresponds well to the 1.4 octaves chosen by Schütt and Wichmann [31]. Note that for Gaussian distributions the standard relationship

between FWHM and standard deviation, applies, so that $w_\theta$ can be converted to $\sigma_\theta$ and vice versa:

$$w_\theta = 2\sqrt{2\log 2}\,\sigma_\theta \tag{4}$$

To model the increase of orientation bandwidth at low SFs, we first modeled the relationship between SF and RF width, which follows a $1/SF$ relationship at SFs above 1 cpd, whereas below 1 cpd RF width increases at a rate of 0.5 [100]. The Gaussian RF spatial constant $\sigma_{RF}$ (as RF width was defined as $2\sigma_{RF}$) can thus also described as a function of spatial frequency $\theta$:

$$\sigma_{RF}(\theta, \beta_0) = \begin{cases} \frac{1}{2}\exp\left(-1.0\log\theta + \log\beta_0\right), & \text{if } \theta \geq 1 \\ \frac{1}{2}\exp\left(-0.5\log\theta + \log\beta_0\right), & \text{if } \theta < 1 \end{cases} \tag{5}$$

where the intercept $\beta_0$ is the RF width estimated at the SF of 1 cpd, which amounts to approximately 1 dva [100]. Second, we estimated the orientation bandwidth of a Gabor filter $w_\omega$ invoking its inherent dependency on its SF and the SD of its Gaussian aperture. Movellan [101] specifies this relationship as

$$w_\omega(\theta, b) = 2\arctan\left(\frac{bC}{\theta}\right) \tag{6}$$

where $C = \sqrt{\frac{\log 2}{\pi}}$ and the variable $b$ has the following relationship to $\sigma_{RF}$:

$$b = \sqrt{\frac{1}{2\pi\sigma_{RF}^2}} \tag{7}$$

Using this model, we could estimate the orientation bandwidth $w_\omega$, and thus also $\sigma_\omega$, of a theoretical RF defined solely based on its SF. Without further assumptions this model's $w_\omega$ closely approximated the orientation bandwidths reported in the literature, that is, 41 deg at SFs above 1 cpd and up to 100 deg at SFs as low as 0.1 cpd (cf. [102], their Fig 1), as well as the orientation bandwidth chosen in [31], that is, ~40 deg. Consequently, to avoid excessively large kernel matrices, we scaled spatial filters to dimensions equaling $8\sigma_{RF}(\theta, 1)$ (see Fig 4a for examples of filter kernels).

Finally, to compute spatial responses to stimulus input $I(t,y,x)$, we convolved each two-dimensional retinal image at a given time point $t$ with every log-Gabor filter kernel in the bank $G(\theta, \omega, \phi)$. This yielded $R_s(\theta, \omega, \phi, t, y, x)$, that is, spatial responses for each time point and for each filter definition.

**5.9.2. Temporal response functions.** While Gabor filters are an adequate model for the spatial response properties of the visual system, a suitable filter kernel is also needed to represent its temporal response properties. As pointed out by Kelly [32], temporal and spatial sensitivity functions are not separable, resulting in a spatiotemporal contrast-sensitivity surface where temporal sensitivity profiles, that describe how sensitivity varies across temporal frequencies (TFs), strongly depend on SF. For 31 SFs (ranging from 0.05 to 10 cpd in logarithmic steps), we therefore derived temporal response functions (TRFs) from flicker sensitivity data, simulating temporal flicker envelopes of 33 TFs (ranging from 0.5 to 50 Hz in logarithmic steps). Since thresholds estimated by flicker could be converted to thresholds estimated by moving stimuli by dividing by an approximate factor of 2 [103], we adapted Kelly's formula ([32], Eq 8) as a ground truth to relate TF to contrast sensitivity for any given SF (for the

adapted formula, see [8]). With this relationship established (Fig 8a), we simulated temporal envelopes of flicker stimuli–one envelope per temporal frequency $f$–with a duration $\delta$ of 7.5 seconds enveloped by a Gaussian contrast ramp with a SD of $\delta/8$ (i.e., 930 ms), thus creating a flicker stimulus well localized in TF space. The temporal envelope is given by:

$$I(t,f) = \sin 2\pi f t \cdot \exp\left(\frac{-(t - \delta/2)^2}{2(\delta/8)^2}\right) \tag{8}$$

These flicker envelopes were convolved with the TRF $H(t)$ of the form

$$H_\theta(t) = A(t/\tau)^n\, e^{-\frac{t}{\tau}}\left(\frac{1}{n!} - B\frac{(t/\tau)^k}{(n+k)!}\right) \tag{9}$$

([104], Eq 4) to estimate the response to the flicker. Then, applying temporal probability summation according to the formula

$$S_\theta(f) = \left[\int_0^t |I(t,f) * H_\theta(t)|^\beta\, dt\right]^{\frac{1}{\beta}} \tag{10}$$

(cf. [104], Eq 3) we computed sensitivity $S$ for a given TF. To estimate parameters of $H_\theta(t)$ for a given SF, we fitted resulting $S(f)$ to the corresponding sensitivity values estimated in [32]. In other words, one SF-specific TRF had to be able to predict contrast sensitivity to the entire TF range. Fig 8b shows the SF-specific TRFs estimated by the procedure described above: Whereas for SFs below 2 cpd, bandpass temporal sensitivity results in biphasic TRFs, TRFs at higher SFs are monophasic, reflecting the lowpass sensitivity profile at these SFs.

To capture all SF-specific TRFs in one model and to be able to extract a TRF for any given SF, we fitted the SF $\times$ time surface (Fig 8c) with a Generalized Additive Model (GAM; [105])

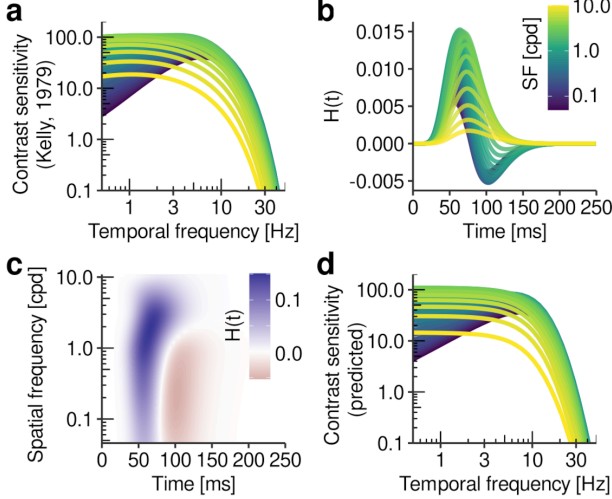

**Fig 8. Estimation of temporal response functions (TRFs). a** Temporal sensitivity functions for a range of SFs (based on [32]). **b** SF-specific TRFs estimated by fitting $H(t)$ parameters to flicker sensitivity data shown in panel a (see text for details). **c** Time-SF surface created by fitting a GAM to TRFs shown in panel b. **d** Flicker sensitivity predicted by convolving simulated flicker stimuli with TRFs extracted from time-SF surface.

including cubic regression-spline terms for time (50 knots) and log SF (10 knots), as well as their tensor product interaction. To evaluate how well this estimated TRF surface could mimic human flicker sensitivity profiles, we computed sensitivity using fitted TRFs across the entire range of TFs and SFs. There was a striking similarity between ground-truth sensitivity (Fig 8a) and sensitivity predicted by our model (Fig 8d), with mean squared errors amounting to 0.016 log units in the displayed range, suggesting that Kelly's spatiotemporal surface could well be reduced to a set of TRFs.

Finally, for the model at hand, we used estimated TRFs $H(t)_\theta$ (one for each element of $S_\theta$, as predicted by the GAM described above) to convolve spatial responses $R_s(\theta, \omega, \phi, t, y, x)$ along the temporal dimension $t$. Note that each point in $R_s$ contained the filter response given by $G(\theta, \omega, \phi) * I(t, y, x)$, but–when analyzed as a function of time–these responses could contain TFs unresolvable by the visual system (for a similar model, see [8]). Convolution with SF-matched TRFs thus constituted a filter to restrict spatial responses to the human visual system's usable TF range, yielding the spatiotemporal response matrix $R_{st}(\theta, \omega, \phi, t, y, x)$. Note that, because flicker sensitivity was measured under retinal stabilization [32], TRFs were not confounded by the effects of eye movements and thus ideal to study the spatiotemporal consequences of saccade-induced retinal motion.

**5.9.3. Squaring and normalization.** As an energy output mechanism [106,107], we squared spatiotemporal responses of even and odd phases

$$R_E(\theta, \omega, t, y, x) = \sqrt{R_{st}(\theta, \omega, \phi_0, t, y, x)^2 + R_{st}(\theta, \omega, \phi_{90}, t, y, x)^2} \tag{11}$$

thus dropping the $\phi$ dimension. Squaring also constrained response magnitude to positive numbers, allowing us to perform normalization as shown below.

Normalization was performed using the delayed normalization model [108], which has proven suitable to describe the temporal dynamics of early visual cortex [33]. The model can describe transient and sustained components of visual responses by performing divisive normalization by temporally lowpass-filtered responses. This temporal lowpass filter takes the form of

$$H_L(t, \tau) = e^{-t/\tau} \tag{12}$$

([33], Eq 3) where $\tau = 0.75$. Since our model was also resolved in space, we implemented an additional spatial lowpass filter, that is, a Gaussian kernel $G_L$ with SD=0.5 dva. Divisive normalization was then performed according to the formula

$$R_N = \frac{R_E^n}{\sigma^n + R_{LP}^n} \tag{13}$$

where normalization constant and exponent were set to $\sigma = 0.07$ and $n = 1.4$, respectively ([33], their Fig 10). Importantly, the normalization denominator $R_{LP}$ was computed from a normalization pool of surrounding SF and orientation cells (±2 adjacent cells), which were weighted by distance using the 2D Gaussian function

$$w_P(\theta, \omega) = 1 - \exp\left(-\left(\frac{\theta}{\sigma_{P,\theta}}^2 - \frac{\omega}{\sigma_{P,\omega}}^2\right)\right) \tag{14}$$

where $\sigma_{p,\theta} = 1$ octave and $\sigma_{p,\omega} = 11.5\,$deg (for parameter selection, see [31]). Pooling was performed with the weighted average function

$$R_{LP}(\theta,\omega) = \frac{\sum_{i_\theta \in S_\theta} \sum_{i_\omega \in S_\omega} R_L(i_\theta, i_\omega) w_P(\theta - i_\theta, \omega - i_\omega)}{\sum_{i_\theta \in S_\theta} \sum_{i_\omega \in S_\omega} w_P(\theta - i_\theta, \omega - i_\omega)} \tag{15}$$

where $R_L$ is the outcome of convolving $R_E$ with temporal and spatial lowpass filters $H_L$ and $G_L$. This convolution, as well as the normalization described above were performed separately for each $\theta \in S_\theta$ and $\omega \in S_\omega$ (omitted above for simplicity). Note that this normalization allowed for a more realistic representation of visual responses to longer stimulation–for instance, at saccade on sets and offsets when targets were largely stable on the retina–whereas mere convolution by spatial and temporal response functions without delayed normalization would result in highly unrealistic, unaltered sustained visual responses. In contrast, transient responses–occurring predominantly when the target stimulus is rapidly shifted through a RF during saccades–were largely unaffected by normalization (see also [33]). The final output of the model reported throughout this manuscript was matrix $R_N$, which resolved visual responses in space and time, as well as in SF and orientation.

**5.9.4. The switch model.** The switch model (see Fig 7a) proposes that the system has a learned internal representation of the typical visual consequences of its own saccades–the sensorimotor contingency $\bar{x}$–that we estimated by averaging all trials' $R_N$ in the static condition (collapsed across SF and orientation channels), separately for each participant. From this time-resolved representation the prediction $\Delta\bar{x}_k^+$ was extracted, that is, by first differentiating $\bar{x}$ and then spatially pooling the maximum positive responses within a temporal integration window $[k - w_i, k + w_i]$, where $w_i = 15$ ms. The prediction thus represents–with relatively low temporal precision–the visual change that would be compatible with a retinal shift (solely) induced by an eye movement. This prediction is compared to the actual visual change $\Delta z_k^+$ received by the system, resulting in the prediction error

$$y_k = \Delta z_k^+ - \Delta x_k^+ \tag{16}$$

that signals a significant deviation from the expected visual change when crossing a detection threshold $\lambda$. This threshold, which serves to suppress small prediction errors that occur very frequently, is determined by the median-based deviation of positive visual change in $\bar{x}$, thus representing the baseline level of visual responses during a saccade. The model updates its estimate of target position $x_k$ by incorporating either the predicted or the measured visual change according to

$$x_{k+1} = x_k + \Delta x_k^+ + K \cdot y_k \tag{17}$$

where $K$ represents the switch:

$$K = \begin{cases} 0, & \text{if } y_k \leq \lambda \\ 1, & \text{if } y_k > \lambda \end{cases} \tag{18}$$

Thus, as soon as a significant prediction error is detected, the model switches from its default mode of relying on its internal prediction to actively tracking (the unpredicted) visual change.

Model simulations (as shown in Fig 4b and 4c, as well as in Fig 7b and 7c and 7d and 7e) were performed on 1000 randomly selected trials, where 10 trials were selected per participant and experimental cell (streak presence × target movement direction).

## Supporting information

### Co-registration of EEG and eye tracking

**S1 Fig. Quality of temporal synchronization of EEG and eye-tracking data. a** Distribution of trigger sample lags when temporally aligning EEG and eye-tracking data after each signal was downsampled to 1000 Hz. **b** Results of cross-correlating EOG and eye-tracking eye position estimates (according to [20]) to check the alignment of both recordings. Positive lags would indicate that eye-movement events occur earlier in EOG than in eye-tracking data. In both panels, individual observers are shown as colored lines and solid black lines represent grand averages.
(TIFF)

**S2 Fig. Effect of correcting ocular artifacts in EEG data, using the OPTICAT procedure** [84]. Panels show each observer's data (all channels, except for EOG and mastoid electrodes) around the time of leftward (upper) and rightward (lower) primary saccades, before and after rejecting critical ICA components (see Sect 5.7.2).
(TIFF)

**S3 Fig. Effect of correcting ocular artifacts in EEG data (conventions same as Fig 5.9.4) around the time of secondary saccades to the target, aggregated across observers.**
(TIFF)

**S4 Fig.  Post-saccadic evoked potentials.** Average fixation-related potentials recorded from electrode Oz ($\pm 1$ *SEM*), or $\lambda$ waves, are shown for all experimental conditions (columns: target motion direction; rows: primary saccade direction $\times$ continuous target motion) and for three secondary saccade-latency bins, each containing the same number of trials in a given experimental cell. Vertical shaded areas indicate each condition's and bin's mean onset of the secondary saccade $\pm 1$ *SEM*.
(TIFF)

### Primary saccade metrics and stimulus timing

**S5 Fig. Distributions of primary saccade metrics (a–c) and stimulus timings (d–e), including both rightward and leftward saccades.** Individual observers are color-coded.
(TIFF)

**S6 Fig. Distributions of secondary saccade latencies (both rightward and leftward primary saccades) for each target motion direction and observer.** Individual observers are color-coded.
(TIFF)

**S7 Fig. Primary saccade metrics.** Left boxplots and circle dots indicate the motion-absent condition, whereas right boxplots and triangle dots indicate the motion-present condition.
(TIFF)

**S8 Fig. Horizontal and vertical components of primary saccade amplitude.** Dotted lines represent instructed saccade amplitude whereas semi-transparent colored crosses indicate the mean amplitude component on each dimension ($\pm 1$ SD) for individual observers in the experimental conditions target motion direction (columns) and continuous motion (rows). Black crosses show their respective population mean $\pm 1$ SEM.
(TIFF)

### Decoding control analyses

**S9 Fig. Generalization of classification results from absent to present motion conditions.** Classifiers were trained on data in absent conditions and then subsequently used to predict motion direction in absent and present conditions. See Fig 2 for figure conventions. (TIFF)

**S10 Fig. EEG decoding results for each target motion direction.** Same as Fig 2b and 2c, only that all analyses were performed relative target motion offset (vertical dotted line at 0 ms), instead of saccade offset. (TIFF)

**S11 Fig. EEG decoding results for each target motion direction and separately for present and absent conditions after removing trials with secondary saccade latencies smaller than 100 ms (upper row), compared to the original results achieved without any latency-based exclusion (lower row; same as Fig 2b).** First-row asterisks denote significance levels of cluster-based permutation tests comparing present and absent conditions with <.05 (first row) and <.01 (second row). (TIFF)

**S12 Fig. EEG decoding results relative to the onset of secondary saccades. a** Distribution of primary saccade offsets. Shades represent different observers. **b** Accuracy of classifying target motion direction, separately for present and absent conditions. Asterisks indicate results of cluster-based permutation tests
comparing these two conditions. **c** Accuracy of classifying present vs absent motion conditions. Cluster-based permutation tests compared classification performance for real and scrambled class labels. All shaded error bars indicate ±1 SEM. First-row asterisks denote a significance level of <.05, whereas second-row denote <.01. Dashed lines show baseline classification accuracy, computed by performing the decoding procedure with scrambled class labels. (TIFF)

## Acknowledgments

We acknowledge the help of Arne Stein, Carmen Haake, Ezgi Temel, Doruk Yiğit Erigüç, Mara Doering, Maren Eberle, and Tobias Richter in supporting EEG data collection, and thank Olaf Dimigen for his valuable remarks on EEG data analysis.

## Author contributions

**Conceptualization:** Richard Schweitzer, Martin Rolfs.

**Data curation:** Richard Schweitzer.

**Formal analysis:** Richard Schweitzer.

**Funding acquisition:** Richard Schweitzer, Jörg Raisch, Martin Rolfs.

**Investigation:** Richard Schweitzer.

**Methodology:** Richard Schweitzer.

**Project administration:** Richard Schweitzer.

**Resources:** Richard Schweitzer.

**Software:** Richard Schweitzer.

**Supervision:** Thomas Seel, Jörg Raisch, Martin Rolfs.

**Validation:** Richard Schweitzer.

**Visualization:** Richard Schweitzer.

**Writing – original draft:** Richard Schweitzer.

**Writing – review & editing:** Richard Schweitzer, Thomas Seel, Jörg Raisch, Martin Rolfs.

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
