## [Decision Letter · Decision Letter 0]

2 Jul 2025

PCOMPBIOL-D-25-00586

Early visual signatures and benefits of intra-saccadic motion streaks

PLOS Computational Biology

Dear Dr. Schweitzer,

Thank you for submitting your manuscript to PLOS Computational Biology. After careful consideration, we feel that it has merit but does not fully meet PLOS Computational Biology's publication criteria as it currently stands. Therefore, we invite you to submit a revised version of the manuscript that addresses the points raised during the review process.

Please submit your revised manuscript within 60 days Sep 01 2025 11:59PM. If you will need more time than this to complete your revisions, please reply to this message or contact the journal office at ploscompbiol@plos.org. Please include the following items when submitting your revised manuscript:

We look forward to receiving your revised manuscript.

Kind regards,

Tim Christian Kietzmann, Dr. rer. nat.

Academic Editor

PLOS Computational Biology

Lyle Graham

Section Editor

PLOS Computational Biology

**Journal Requirements:**

3) Your manuscript is missing the following sections: Abstract.  Please ensure all required sections are present and in the correct order. Make sure section heading levels are clearly indicated in the manuscript text, and limit sub-sections to 3 heading levels. An outline of the required sections can be consulted in our submission guidelines here:

Potential Copyright Issues:

i) Please confirm (a) that you are the photographer of 1A, 1B, and 4A, or (b) provide written permission from the photographer to publish the photo(s) under our CC BY 4.0 license.

2) If any authors received a salary from any of your funders, please state which authors and which funders..

7) Your current Financial Disclosure states, "Yes ↳ Please add funding details. R.S., M.R., and J.R. were funded by the Deutsche Forschungsgemeinschaft (DFG, German Research Foundation) under Germany’s Excellence Strategy – EXC 2002/1 ”Science of Intelligence” – project number 390523135. R.S. was supported by the Studienstiftung des deutschen Volkes during the early stages of thestudy, as well as by the Italian Ministry for Universities and Research (MUR) and the European Union within the Next Generation EU framework (grant ’T-GAZE’, CUP E73C22000480001) in the final stages of the study. M.R. has received funding from the European Research Council (ERC) under the European Union’s Horizon 2020 research and innovation programme (grant agreement No 865715) as well as from the Heisenberg Programme of the DFG (grants RO3579/8-1 and RO3579/12-1). ↳ Please select the country of your main research funder (please select carefully as in some cases this is used in fee calculation). GERMANY - DE".

However, your funding information on the submission form indicates missing for Italian Ministry for Universities and Research (MUR)

and European Union (Next Generation EU framework, grant ’T-GAZE’). 

Please indicate by return email the full and correct funding information for your study and confirm the order in which funding contributions should appear. Please be sure to indicate whether the funders played any role in the study design, data collection and analysis, decision to publish, or preparation of the manuscript.

8) Please ensure that the funders and grant numbers match between the Financial Disclosure field and the Funding Information tab in your submission form. Note that the funders must be provided in the same order in both places as well. Currently, the Financial Disclosure states there was no funding received.

**Reviewers' comments:**

Reviewer's Responses to Questions

**Comments to the Authors:**

Reviewer #1: Thank you for the opportunity to provide a review of the manuscript entitled “Early visual signatures and benefits of intra-saccadic motion streaks” by Schweitzer and colleagues.

In the manuscript, the authors report an interesting combined EEG/eyetracking study in which a high-temporal resolution display was used to tightly control stimulus presentation during a saccade, whilst the eyes were in motion. By moving the saccade target either smoothly or abruptly (after a 25ms delay), the authors elicited secondary saccades, and compared the properties of those secondary saccades across smooth/jump conditions. In addition, they used EEG decoding to investigate how neural signatures of target position evolved during this period.

Overall, the manuscript is well-written, with a well-designed experiment and a strong theoretical underpinning. I am overall sympathetic to the conclusions drawn from the study, and the results neatly fit with previous work by the same authors and complement those earlier conclusions.

My main concern is a methodological issue related to the interpretation of one of the EEG analysis approaches. Specifically, the authors train classifiers to decode the position of the final target position (1 out of 5 possible positions) and, then test this on saccade trials, contrasting a smooth movement condition where the target shifts smoothly to the new location vs a “jump” condition where the target abruptly disappears and reappears in the new location.

This introduces a confound for the unfortunate reason that during the 25 ms gap, there is contrast energy on the screen in one condition that is not there in the other condition. Importantly, the location of that contrast energy (naturally) directly corresponds to the future location of the reappearing object (since it is moving on its way there). As such, it is unsurprising that it is systematically informative about the future position of the object – not necessarily because that information is used in a neural computation, but merely because those trials provide earlier visual input that “looks like” the to-be-decoded target location. The finding that decoding performance ramps up earlier in the smooth motion condition (e.g. Fig 7b) is therefore unsurprising – this is expected, simply because there is information on the screen earlier in that condition, which is visually (and therefore neurally) similar to what the classifier is trained to discriminate (position).

In the jump condition, there is no visual information available until reappearance, so decoding performance naturally cannot rise. I therefore don’t think we can really draw an important conclusion from this result.

Ultimately I don’t think this is a huge problem, because the more interesting finding (to me) is that the orientation decoding works – even generalizing across motion directions (and therefore starting and ending positions). Although the same problem holds (there is something informative on the screen in one condition before anything becomes available in the other condition), the fact that it generalizes across direction is, to me, a stronger demonstration that the streak is indeed encoded as orientation, and that that information is available to be used to accelerate subsequent processing.

Minor point: the paper is overall well-written, but I note that the paper seems to almost deliberately avoid the topic of saccadic remapping. I understand that remapping is distinct from intrasaccadic perception, but particularly given the methodological approach of using EEG decoding during this period, it seems relevant to discuss recent work using similar approaches, including Fabius et al in JNeuro (10.1523/JNEUROSCI.1169-20.2020) and Moran et al JNeuro (10.1523/JNEUROSCI.2134-23.2024; particularly relevant because position was decoded, like in the current manuscript).

Reviewer #2: This is an excellent study, using concurrent EEG and eye tracking to examine the perception of continuous motion during saccadic eye movements. Although previous behavioral findings from this group are already quite compelling, the present study adds to these findings by suggesting that the source of the effect lies in the visual cortex and offers more detailed temporal information on its dynamics. Then, they develop a computational model that sheds new light on the physiological correlates of this perceptual phenomenon. The model supports the hypothesis that the target’s motion trajectory is coded by orientation-selective circuits.

The study is well designed and includes thorough, state-of-the-art analyses that are, without doubt, well thought out. The findings are interesting and robust. I support the publication of this paper, but I also have a few concerns that should be addressed before it is published.

First, a general comment on the writing of the paper. The manuscript is highly impressive in terms of the analyses, the computational model, and the depth of thought evident in every detail. However, the writing could be clearer and more communicative. I found it very difficult to navigate the paper and locate answers to some of my questions. Even though the study overlaps significantly with my own field of expertise, I found it difficult to read and understand. I’m concerned that, for the broader readership of PLOS Computational Biology, this would be an almost impossible task.

My second concern is methodological. The authors argue that they find a significant difference in classification performance between the two conditions (absent and continuous motion) during the short period between the offset of the first saccade and the onset of the second. The offset of the first saccade is well defined per trial (as the classification is segmented relative to this offset), but the timing of the second saccade is not well considered. The authors claim that the effect exists prior to the onset of the second saccade, but in fact, they did not check the second saccade onset in individual trials and (as far as I understand) not even in individual participants. Instead, they rely on the average time of the second saccade across participants. Since saccade onsets are highly variable and follow an ex-Gaussian distribution, it is likely that the earliest secondary saccades (which fall at the beginning of the distribution’s rising slope) occur much earlier than the average.

This is especially problematic because the oculomotor and visual activity that accompanies saccades in the EEG is highly correlated with the direction of the saccade. Classification is highly sensitive to informative data and can therefore be significantly affected by even a small number of saccades. In other words, even a few secondary saccades that occur earlier than the rest may aid classification and lead to misinterpretation. Moreover, the findings show a behavioral difference in saccade onsets between the conditions, with real motion occurring earlier than apparent motion. This again raises the question of whether the EEG effects stem from differences in independent visual activity or are the direct result of eye movements.

A possible and (relatively) easy solution could be to exclude all trials with saccades in the early time range (e.g., until 100 ms), and then focus the analysis only on that window.

I also have a few minor comments:

1. I think it is important to show not only the decoding outcome but also the ERP (the lambda wave).

2. Page 2, line 81: missing “of the trials” or “of the cases.”

3. Page 2, line 92: saccade latencies — are they calculated relative to the first saccade offset or to stimulus onset?

**Have the authors made all data and (if applicable) computational code underlying the findings in their manuscript fully available?**

Reviewer #1: Yes

Reviewer #2: None

PLOS authors have the option to publish the peer review history of their article (what does this mean?). If published, this will include your full peer review and any attached files.

Reviewer #1: No

Reviewer #2: No

**Figure resubmission:**
---

## [Decision Letter · Decision Letter 1]

22 Sep 2025

Dear Dr. Schweitzer,

We are pleased to inform you that your manuscript 'Early visual signatures and benefits of intra-saccadic motion streaks' has been provisionally accepted for publication in PLOS Computational Biology.

Best regards,

Tim Christian Kietzmann, Dr. rer. nat.

Academic Editor

PLOS Computational Biology

Lyle Graham

Section Editor

PLOS Computational Biology

Reviewer #1:

Reviewer #2:

Reviewer's Responses to Questions

**Comments to the Authors:**

Reviewer #1: Dear Editor,

Thank you for the opportunity to review the revised version of this manuscript.

The authors have done a great job revising the manuscript in line with my comments and those of the other reviewer.

I apologise for having missed the additional analysis in Appendix C, but appreciate the authors revising the paper to include more discussion of this issue in the main text.

I have no further comments.

Reviewer #2: The authors have carefully revised their manuscript to address all my comments. I believe it is now ready for publication. Good luck

**Have the authors made all data and (if applicable) computational code underlying the findings in their manuscript fully available?**

Reviewer #1: Yes

Reviewer #2: Yes

PLOS authors have the option to publish the peer review history of their article (what does this mean?). If published, this will include your full peer review and any attached files.

Reviewer #1: No

Reviewer #2: No

---

## [Editor Report · Acceptance letter]

PCOMPBIOL-D-25-00586R1

Early visual signatures and benefits of intra-saccadic motion streaks

Dear Dr Schweitzer,

I am pleased to inform you that your manuscript has been formally accepted for publication in PLOS Computational Biology. Your manuscript is now with our production department and you will be notified of the publication date in due course.

With kind regards,

Zsofia Freund
